

# Shallow water memory: Stokes and Darwin drifts

**M. M. Sheikh-Jabbari**[1*], **V. Taghiloo**[1,2†] **and M. H. Vahidinia**[1,2‡]

**1** School of Physics, Institute for Research in Fundamental Sciences (IPM),
P.O.Box 19395-5531, Tehran, Iran
**2** Department of Physics, Institute for Advanced Studies in Basic Sciences (IASBS),
P.O. Box 45137-66731, Zanjan, Iran

⋆ jabbari@theory.ipm.ac.ir , † vahidtaghiloo@ipm.ir , ‡ vahidinia@iasbs.ac.ir

## Abstract

It has been shown in [1] that shallow water in the Euler description admits a dual gauge theory formulation. We show in the Lagrange description this gauge symmetry is a manifestation of the 2 dimensional area-preserving diffeomorphisms. We find surface charges associated with the gauge symmetry and their algebra, and study their physics in the shallow water system. In particular, we provide a reinterpretation of the Kelvin circulation theorem in terms of conserved charges. In the linear shallow water case, the charges form a u(1) current algebra with level proportional to the Coriolis parameter over the height of the fluid. We also study memory effect for the gauge theory description of the linearized shallow water and show Euler, Stokes and Darwin drifts can be understood as a memory effect and/or change of the surface charges in the gauge theory description.



# 1 Introduction

Shallow water systems are fluid systems with a much larger extent in two dimensions than the third. The atmosphere and oceans are typical examples of shallow fluid systems [2, 3]. See [4] for a recent and nice set of lecture notes on fluid mechanics. In a shallow water system, the dynamics takes place in the surface of the fluid, surface waves, and we are hence dealing with an effective 2+1 dimensional system. We usually take the 3 dimensional fluid to be incompressible. This provides a continuity/conservation equation. There is also Kelvin circulation theorem [2, 3], stating that an irrotational fluid remains irrotational. This yields conservation of circulation, the integral of vorticity over a two-dimensional surface. These two conservation equations, upon Noether's theorem, may be associated with symmetries of the system. In an interesting recent paper [1], it was argued that these two conservation equations may be naturally described by a $2+1$ dimensional $u(1) \times u(1)$ gauge theory. In this gauge theory description Kelvin's theorem corresponds to the Gauss law.

Shallow water system is governed by nonlinear equations but is well approximated by a linear system when amplitude of the waves is much smaller than the height of the fluid. The linear system is described by a 2+1 dimensional Maxwell-Chern-Simons (MCS) gauge theory [1]. The study in [1] was motivated by bridging between the two dimensional condensed matter systems, in particular quantum Hall system, and the shallow water mechanics focusing on the topological features of the two sides (for further discussion see [5, 6]).

Here we explore what more can one learn from the gauge theoretic description of the shallow water system. It is well-known that in gauge theories there exists an infinite set of surface charges, which may be viewed as a local extension of the usual global charges defined by the Gauss law, see e.g. [7]. In the fluid case, as we show in section 3, we argue how the gauge theory description yields an infinite set of conserved charges for the fluid, providing a local extension of Kelvin's circulation theorem. In the electromagnetic (as well as in gravitational systems) the change in these surface charges due to passage of a wave gives rise to the memory effects, see e.g. [7–12]. We study a similar memory effect for MCS theory describing the linearized shallow water system and show how various drifts discussed in fluid mechanics [2, 3] can be understood as MCS memory effects.

**Outline of the paper.** In section 2 we review gauge theoretic description of the Euler formulation of nonlinear and linear shallow water system. In section 3 we study symmetries and surface conserved charges of the $u(1) \times u(1)$ nonlinear theory and the linearized MCS theory. In section 4 we provide Lagrange formulation of the shallow water system and show that the gauge symmetry of the linearized MCS in the Euler description is related to area preserving diffeomorphisms in the Lagrange formulation. In section 5 we discuss memory effect in the linearized gauge theory description. In section 6 we discuss Stokes drift [13–15] and how it is realized in the gauge theory description. In section 7 we consider shallow water system in an external force and compute Darwin drift [16, 17] as a memory effect. Section 8 is devoted to a brief summary and outlook. In appendix A we present derivation of the Green's function and in B we briefly discuss memory effect in nonlinear shallow water.

## 2 Shallow water gauge theory: A review

Nonlinear shallow water system is described by two dynamical fields in $2 + 1$ dimensions: $\mathcal{H}(x^i, t)$ which describes the height of the fluid and $u^i(x^j, t)$; $i, j = 1, 2$ the horizontal velocity of the fluid. In the Euler description, dynamics of this system is governed by

$$\frac{D\mathcal{H}}{Dt} := \frac{\partial \mathcal{H}}{\partial t} + u \cdot \nabla \mathcal{H} = -\mathcal{H} \nabla \cdot u, \tag{1a}$$

$$\frac{Du_i}{Dt} := \frac{\partial u^i}{\partial t} + (u \cdot \nabla) u^i = f \epsilon_{ij} u^j - g \partial_i \mathcal{H}. \tag{1b}$$

The first equation (1a) is the mass conservation equation, that the 3 dimensional fluid is incompressible and (1b) is Newton's second law (Navier-Stokes or Euler equation), in which $g$ is the gravitational constant and $f$ is the Coriolis parameter, which is a constant.[1] As we see these equations are nonlinear for both variables $\mathcal{H}, u^i$.

This theory has two global Noether currents which satisfy

$$\partial_t \mathcal{H} + \nabla \cdot (\mathcal{H} u) = 0, \tag{2a}$$

$$\partial_t (\zeta + f) + \nabla \cdot [(\zeta + f) u] = 0, \tag{2b}$$

where $\zeta = \epsilon^{ij} \partial_i u_j$ is the vorticity of the shallow water. Eq.(2a) is the same as (1a), the incompressibility of the 3 dimensional fluid and hence the associated conserved charge is mass (or mass density). Eq. (2b) arises from (1b) and its associated conserved Noether charge is

$$\Gamma_0 := \int_\Sigma d^2 x \, (\zeta + f) = \oint_\mathcal{B} dl_i \left( u^i - \frac{f}{2} \epsilon^{ij} x_j \right), \tag{3}$$

where $\Sigma$ is a constant time slice (Cauchy surface), $\mathcal{B} = \partial \Sigma$ is its boundary and $dl_i$ is the length element along $\mathcal{B}$. In the second equation, we used Stokes' theorem. Circulation $\Gamma_\mathcal{C}$ which is subject to Kelvin's circulation theorem [2, 3] is a generalization of $\Gamma_0$ as,

$$\Gamma_\mathcal{C} := \int_\mathcal{S} d^2 x \, (\zeta + f) = \oint_\mathcal{C} dl_i \left( u^i - \frac{f}{2} \epsilon^{ij} x_j \right), \tag{4}$$

where as depicted in Fig. 1, $\mathcal{S}$ is a generic section of $\Sigma$ and $\mathcal{C} = \partial \mathcal{S}$ is its boundary. That is, $\mathcal{C}$ a generic closed path in the fluid. The fact that $\Gamma_\mathcal{C}$ can be written as a codimension 2 integral suggests that this system should have a gauge theory description. Indeed as shown in section 3 that $\Gamma_\mathcal{C}$ appears as a conserved charge.

---

[1] Here we take $f$ to be a constant. However, for the system of oceans on the Earth, the Coriolis parameter $f$ depends on the latitude $\theta$, $g \simeq 9.8 m/sec^2$ and $f = 1.45 \times 10^{-5} \sin \theta / sec$. So, for Equator $f$ is very small and is the largest near the poles. $f$ is positive (negative) in the northern (southern) hemisphere.

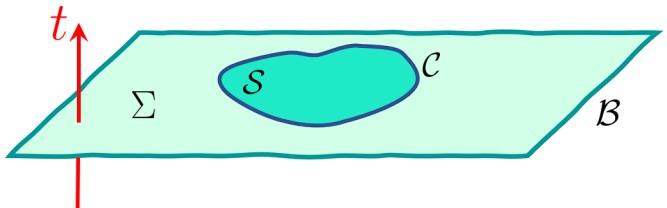

Figure 1: $\Sigma$ is a constant time slice, a Cauchy surface, $\mathcal{B}$ is its boundary. $\mathcal{S}$ is a generic section of $\Sigma$ and $\mathcal{C}$ is its boundary.

## 2.1 Nonlinear gauge theory

One may view (2) as Bianchi identities of a 2+1 dimensional $u(1) \times u(1)$ gauge theory. In this sense, this gauge theory provides a (Hodge) dual description of the fluid. Explicitly, let the electric field of the two gauge theories be denoted by $E_i, \tilde{E}_i$ and the corresponding magnetic fields by $B, \tilde{B}$. The corresponding Bianchi identities are $\partial_t B - \epsilon^{ij} \partial_i E_j = 0$ and similarly for the tilde gauge field. Therefore, upon identifications, see [1] for more details,

$$
\begin{aligned}
B &= \mathcal{H}, & E_i &= \epsilon_{ij} \mathcal{H} u^j, \\
\tilde{B} &= f + \zeta, & \tilde{E}_i &= \epsilon_{ij} u^j (f + \zeta).
\end{aligned}
\tag{5}
$$

Bianchi identities recover (2). Given the Bianchi identities, one may try to obtain (1) as the equation of motion (EoM) of this gauge theory. This has been recently studied by David Tong [1], where the following action was introduced[2]

$$
S[A_\mu, \alpha, \beta] = \int dt\, d^2x \left( \frac{E^2}{2B} - \frac{1}{2} g B^2 + f A_0 - \epsilon^{\mu\nu\rho} A_\mu \partial_\nu \beta \partial_\rho \alpha \right),
\tag{6}
$$

where $\mu = 0, 1, 2$ and $E_i := \partial_t A_i - \partial_i A_0$ and $B := \epsilon^{ij} \partial_i A_j$ are electric and magnetic fields associated with gauge field $A_\mu$ respectively. One can use Clebsch parametrization and rewrite the two scalars $\alpha$ and $\beta$ in terms of a new dummy gauge field [1]

$$
\begin{aligned}
\tilde{A}_\mu &= \partial_\mu \chi + \beta \partial_\mu \alpha, & \tilde{F}_{\mu\nu} &= \partial_\mu \tilde{A}_\nu - \partial_\nu \tilde{A}_\mu, \\
\tilde{E}_i &:= \tilde{F}_{0i}, & \tilde{F}_{ij} &:= \epsilon_{ij} \tilde{B}.
\end{aligned}
\tag{7}
$$

In terms of $\tilde{A}_\mu$ the last term in (6) takes the form of a $u(1) \times u(1)$ Chern-Simons term $\epsilon^{\mu\nu\rho} A_\mu \partial_\nu \tilde{A}_\rho$.

This action is invariant (up to boundary terms) under the gauge transformations $A_\mu \to A_\mu + \partial_\mu \Lambda$ and $\tilde{A}_\mu + \partial_\mu \tilde{\Lambda}$, where $\Lambda$ and $\tilde{\Lambda}$ are our gauge parameters which are arbitrary functions on spacetime. One may check that EoM for the gauge field $A_\mu$ yields

$$
\mathcal{E}^i = -\partial_t \left( \frac{E^i}{B} \right) - \epsilon^{ij} \partial_j \left( \frac{E^2}{2B^2} + g B \right) + \epsilon^{ij} \tilde{E}_j = 0.
\tag{8}
$$

Using (5) it is straightforward to show that this equation implies (1b), while (1a) appears as Bianchi identity for $A_\mu$ gauge field.

---

[2]If we denote dimension of quantity $X$ by $[X]$, with the conventions used, $[f] = T^{-1}, [g] = LT^{-2}$ and $[B] = L, [E_i] = L^2 T^{-1}, [\tilde{B}] = T^{-1}, [\tilde{E}_i] = LT^{-2}$. The action (6) will have the correct dimensions if it is multiplied by the density of the 3 dimensional water, which we have set equal to 1.

## 2.2 Linearized and effective gauge theory

Assuming that the variations of $\mathcal{H}, u^i$ are small, one may linearize the shallow water equations (1). This may be achieved through $\mathcal{H}(x^i, t) = H + \eta(x^i, t)$ and $u(x^i, t) = 0 + u(x^i, t)$, where $H$ is a constant height and $\eta \ll H$.[3] Substituting these quantities in the EoM (1) and keeping only linear terms in $\eta$ and $u$, we get

$$\partial_t \eta + H\nabla \cdot u = 0, \tag{9a}$$

$$\partial_t u_i = f\epsilon_{ij}u^j - g\partial_i\eta. \tag{9b}$$

These linearized EoM yield two global conservation laws which are the linearized version of conservation laws (2a) and (2b)

$$\partial_t \eta + H\nabla \cdot u = 0, \tag{10a}$$

$$\partial_t \zeta + f\nabla \cdot u = 0. \tag{10b}$$

Therefore the potential vorticity,

$$Q := H\zeta - f\eta, \tag{11}$$

is time-independent, $\partial_t Q = 0$, while it can have $x^i$ dependence.[4] So, for the linearized shallow water we have only one independent conservation law, and the other is replaced with $\partial_t Q = 0$. Note also that (10b) can be rewritten as $\nabla \cdot \tilde{u} = 0$, $\tilde{u}_i := u_i + \frac{1}{f}\epsilon_{ij}\partial_t u_j = -\frac{g}{f}\epsilon_{ij}\partial_j\eta$.

In the linearized level (5) takes the form

$$B = H + \eta = \frac{H}{f}(f + \zeta) - \frac{Q}{f}, \qquad E_i = H\epsilon_{ij}u^j, \tag{12}$$
$$\widetilde{B} = f + \zeta, \qquad\qquad\qquad \widetilde{E}_i = f\epsilon_{ij}u^j,$$

where in the second equality for $B$ we used (10) which implies $\eta = (H\zeta - Q)/f$, with $Q$ being time-independent. Therefore, $\widetilde{B} = \frac{f}{H}B + \frac{Q}{H}$, $\widetilde{E}_i = \frac{f}{H}E_i$ and hence $\widetilde{A}_0 = \frac{f}{H}A_0$, $\widetilde{A}_i = \frac{f}{H}A_i - \frac{Q}{2H}\epsilon_{ij}x^j$, up to a gauge transformation $\tilde{\Lambda}$. That is, $\tilde{A}_\mu$ may be solved in terms of $A_\mu$. We finally note that the linearized equations (9) in terms of $E_i, B$ take the form

$$\nabla \cdot E - f(B - H) = Q, \tag{13a}$$

$$\partial_t E_i = \epsilon_{ij}(fE_j - v^2\partial_j B), \qquad v^2 := gH. \tag{13b}$$

Here $v$ is the speed of gravity waves in the shallow water system.

Let us now focus on the $Q = 0$ sector. In this case, $\tilde{A}_\mu = \frac{f}{H}A_\mu$, up to a gauge transformation. Eliminating $\tilde{A}_\mu$ in favor of $A_\mu$ (in more standard field theory terminology, integrating out $\tilde{A}_\mu$), the linearized equations (9) in this sector may be obtained from a three-dimensional Maxwell-Chern-Simon (MCS) action [1],[5]

$$S = -\frac{g}{4v}\int d^3x \left(F^{\mu\nu}F_{\mu\nu} - \frac{f}{v}\epsilon^{\mu\nu\rho}A_\mu F_{\nu\rho}\right), \tag{14}$$

where $F_{\mu\nu} = \partial_\mu A_\nu - \partial_\nu A_\mu$ is the field strength, $F_{0i} = E_i$ with $E_i$ given in (12) and $F_{ij} = \epsilon_{ij}\eta$. We lower and raise the indices with the metric,

$$g_{\mu\nu} = \text{diag}(-v^2, 1, 1), \tag{15}$$

---

[3]For the oceans and seas on the Earth, where height of water $H$ is typically much larger than the typical amplitude of waves and tides $\eta$, the linearized shallow water approximation is a very good one.

[4]While here we have introduced $Q$ in the linear theory, one can extend the notion of potential vorticity to the nonlinear theory [1].

[5]For the generic $Q \neq 0$ case, $Q$ appears as a background electric charge in the MCS theory and the term $\frac{Q}{H}A_0$ should be added to the Lagrangian.

$d^3x = \sqrt{-g}\,dt\,d^2x = v\,dt\,d^2x$, $\epsilon_{\alpha\beta\mu} = \sqrt{-g}[\alpha\beta\mu] = v[\alpha\beta\mu]$ and $\epsilon^{\alpha\beta\mu} = -\frac{1}{v}[\alpha\beta\mu]$, where $[\alpha\beta\mu]$ is purely antisymmetric symbol which takes values 0 or ±1.[6] We note that (14) has a single $u(1)$ gauge symmetry and eliminating the $\tilde{A}_\mu$ gauge field, has led to the Chern-Simons term. We will return to this point later.

EoM of the MCS action (14),

$$E_\nu := g\left(\nabla^\mu F_{\mu\nu} + \frac{f}{2v}\epsilon_{\nu\alpha\beta}F^{\alpha\beta}\right) = 0, \tag{16}$$

by construction, recovers (13) at $Q = 0$. It is well known that due to the presence of the Chern-Simon term in the action (14), under the gauge transformation, $\delta_\Lambda A_\mu = \partial_\mu\Lambda$,

$$\delta_\Lambda S = \frac{f}{4H}\int d^3x\, \partial_\mu\left(\Lambda\epsilon^{\mu\nu\rho}F_{\nu\rho}\right). \tag{17}$$

This boundary term opens the window for the appearance of the edge modes in the presence of the boundary [1, 18–20].

## 2.3 Poincare and Kelvin coastal waves

Eq. (16) describes various wave solutions in the $Q = 0$ sector. For solutions with $Q \neq 0$, see [1]. Among them, here we discuss two most famous classes of solutions, the *Poincare waves* and the *Kelvin coastal waves*.

**Poincare waves [2–4].**    To explore wave solutions we start with the following ansatz,

$$u_i = \hat{u}_i e^{i(\omega t - k \cdot x)}, \qquad \eta = \hat{\eta} e^{i(\omega t - k \cdot x)}. \tag{18}$$

Plugging the above into the linearized shallow water equations (9) we get the eigenvalue problem

$$\begin{pmatrix} 0 & Hk_1 & Hk_2 \\ gk_1 & 0 & -if \\ gk_2 & if & 0 \end{pmatrix}\begin{pmatrix} \hat{\eta} \\ \hat{u}_1 \\ \hat{u}_2 \end{pmatrix} = \omega \begin{pmatrix} \hat{\eta} \\ \hat{u}_1 \\ \hat{u}_2 \end{pmatrix}. \tag{19}$$

By solving this equation, the eigenvalues are given by

$$\omega^2 = v^2 k^2 + f^2, \qquad \text{and} \qquad \omega = 0. \tag{20}$$

The corresponding eigenvectors are

$$\text{For } \omega = 0: \quad \begin{pmatrix} \hat{\eta} \\ \hat{u}_1 \\ \hat{u}_2 \end{pmatrix} = \begin{pmatrix} 1 \\ igk_2/f \\ -igk_1/f \end{pmatrix}. \quad \text{For } \omega \neq 0: \quad \begin{pmatrix} \hat{\eta} \\ \hat{u}_1 \\ \hat{u}_2 \end{pmatrix} = \begin{pmatrix} Hk^2 \\ k_1\omega - if k_2 \\ k_2\omega + if k_1 \end{pmatrix}. \tag{21}$$

Poincare waves ($\omega \neq 0$ modes) are described by the MCS gauge field (in the temporal gauge $A_t = 0$),

$$A_i = \xi_i e^{i(\omega t - k \cdot x)}, \qquad \xi_2 = -\frac{\omega k_1 - if k_2}{\omega k_2 + if k_1}\xi_1 = -\frac{k_1 k_2 v^2 - if\omega}{k_2^2 v^2 + f^2}\xi_1, \tag{22}$$

---

[6]The above action is written in terms of quantities that are more natural for the fluid (shallow water) description. One needs to make some scalings in $A_\mu$ to write it in terms of quantities and units more natural to the usual MCS theory, where the CS coupling is dimensionless in natural units. In such appropriate units, CS coupling is proportional to $f/H$.

which may be written in a more useful form as,

$$A_1 = \frac{h}{\omega k}\sqrt{v^2 k_2^2 + f^2}\cos(\omega t - k \cdot x + \phi_0),$$
$$A_2 = -\frac{h}{\omega k}\sqrt{v^2 k_1^2 + f^2}\cos(\omega t - k \cdot x + \phi_0 - \varphi), \quad \tan\varphi = \frac{f\omega}{k_1 k_2 v^2}, \tag{23}$$

where $h, \phi_0$ are two real $k$-dependent integration constants. Appearance of the $k$-dependent phase $\varphi$ in $A_2$ is a manifestation of the fact that the Coriolis term (the Chern-Simons term) breaks parity (or time reversal) invariance of the system. The dispersion relation for Poincare waves is $\omega^2 = v^2 k^2 + f^2$, $v^2 = gH$, i.e. the phase of these waves travel at the speed $v$ and $f$ appears as an "effective mass". As we see for Poincare waves $\omega \geq f$, Poincare waves have a low-frequency cutoff $f$.

The velocity field $u^i$ in the temporal gauge is given by $u^i = -\frac{1}{H}\epsilon^{ij}\partial_t A_j$ and hence

$$u_1 = -\frac{h}{Hk}\sqrt{v^2 k_1^2 + f^2}\sin(\omega t - k \cdot x + \phi_0 - \varphi),$$
$$u_2 = -\frac{h}{Hk}\sqrt{v^2 k_2^2 + f^2}\sin(\omega t - k \cdot x + \phi_0), \tag{24}$$
$$\eta = -h\sin(\omega t - k \cdot x + \phi_0 - \hat\varphi), \quad \tan\hat\varphi = \frac{k_1 f}{k_2 \omega}.$$

As the above explicitly shows, in our parametrization $h$ is the square root of the average of $\eta^2$ over a period. Note the $k$-dependent phase difference between $u_1, u_2$ and $\eta$, $\varphi, \hat\varphi$, and that this phase difference vanishes in the absence of the Coriolis parameter $f$. The average of the velocity-squared over a period is

$$\bar{u}^2 = \frac{h^2}{2H^2}(\omega^2 + f^2)/k^2 = \frac{h^2}{2H^2}v^2\left(1 + \frac{2}{k^2 R^2}\right), \qquad R := \frac{v}{f}, \tag{25}$$

where $R$ is the Rossby radius of deformation [21]. Energy density of the gravity waves is proportional to $\bar{u}^2$. Therefore, the energy increases as we increase the wave-length. For long wave-length waves (small $k$) the energy of the wave becomes large and for short wave-length (large $k$), the energy becomes $k$-independent and takes its lowest value proportional to $h^2 v^2/H^2$. For typical Poincare waves in oceans on the Earth, $kR \gg 1$ and hence their energy is essentially $k$ independent.

Zero frequency modes of (16) which are time-independent solutions are governed by

$$E_i = vR\partial_i \eta, \qquad (\nabla^2 - R^{-2})\eta = 0, \tag{26}$$

or equivalently,

$$A_i = vR\partial_i \eta\, t + \hat{A}_i, \qquad \left(\nabla^2 - R^{-2}\right)\hat{A}_i = 0, \qquad \eta = \epsilon^{ij}\partial_i \hat{A}_j. \tag{27}$$

The above in the fluid dynamics are called the *geostrophic balance equations*, see e.g. [4]. Zero frequency modes can be formally viewed as imaginary wave length Poincare waves such that $k^2 v^2 + f^2 = 0$ or $k^2 R^2 + 1 = 0$.

**Coastal Kelvin waves [4,21,22].** Assume we have a boundary at $x_1 = 0$ such that the fluid exists only in the $x_1 > 0$ and for $x_1 < 0$, there is only land. We assume $u_1|_{x_1=0} = 0$, ensuring that no flow passes the boundary. Let us for simplicity explore solutions that have $u_1 = 0$ everywhere. Then, the linearized shallow water equations (9) reduce to

$$\partial_t \eta + H\partial_2 u_2 = 0, \qquad \partial_t u_2 + g\partial_2 \eta = 0, \qquad u_2 = \frac{g}{f}\partial_1 \eta, \tag{28}$$

which yield

$$u_2 = -\frac{h}{H}\, v\, e^{-\frac{x_1}{R}} \cos(k(x_2 + vt) + \phi_0), \qquad \eta = h\, e^{-\frac{x_1}{R}} \cos(k(x_2 + vt) + \phi_0), \qquad (29)$$

where $h$ is the amplitude of the wave and $\phi_0$ is an initial phase and both are in general $k$-dependent constants. Note that to get a wave which falls off in $x_1 > 0$ region, one should choose left moving waves in $x_2$ direction. (The left moving waves in $x_2$ direction are exponentially damping in the $x_1 < 0$ region.) Kelvin waves may be viewed as a certain class of Poincare modes with imaginary $k_1$, $v^2 k_1^2 + f^2 = 0$ and $\omega = v k_2$.

The square root of the average of $\eta^2$ over a period $\bar{\eta} = h\, e^{-x_1/R}$ and the average of the velocity-squared over a period is $\bar{u}^2 = \frac{h^2}{2H^2} v^2 e^{-2x_1/R}$, which is $k$-independent for a given $h$. The value of $\bar{\eta}$ and $\bar{u}^2$ have their maximum values at the boundary $x_1 = 0$. The coastal modes, as the above explicitly shows, describe one dimensional modes, i.e. edge modes.[7] Curiously, the energy of a Kelvin wave of a given amplitude is independent of the frequency.

The most general solution is then given by

$$u_1 = 0, \qquad u_2 = e^{-\frac{x_1}{R}} U(x_2 + vt), \qquad \eta = -\frac{v}{g} e^{-\frac{x_1}{R}} U(x_2 + vt). \qquad (30)$$

For this solution, circulation is $\zeta = -\frac{1}{R} e^{-\frac{x_1}{R}} U(x_2 + vt)$ and in the equivalent gauge field description it is given by,

$$E_1 = H e^{-\frac{x_1}{R}} U(x_2 + vt), \qquad E_2 = 0, \qquad B = -\frac{1}{v} E_1, \qquad (31)$$

or

$$A_t = 0, \qquad A_1 = H e^{-\frac{x_1}{R}} \int^t ds\, U(x_2 + vs), \qquad A_2 = 0. \qquad (32)$$

# 3 Symmetry and conserved charges

The existence of an action principle for the nonlinear shallow water equation allows us to systematically study the symmetries and their associated conserved charges by the machinery of the Noether theorem. In this section, we study the symmetries and associated conserved charges of the shallow water by using actions (6) and (14). We only focus on the gauge symmetry on the gauge field $A_\mu$, while $\tilde{A}_\mu$ gauge field is viewed as a dummy field introduced to produce EoM (through Clebsch parametrization).

## 3.1 Nonlinear theory

The standard machinery of the Noether theorem yields the following Noether current for the gauge symmetry of (6)

$$\rho_\Lambda = \partial_i \left( \Lambda \epsilon^{ij} u_j \right),$$
$$J_\Lambda^i = -\epsilon^{ij} \left[ u_j \partial_t \Lambda + \left( \frac{u^2}{2} + gh \right) \partial_j \Lambda \right] + \Lambda(f + \zeta) u^i. \qquad (33)$$

It is straightforward to show that $\partial_\mu J_\Lambda^\mu = -\partial_i(\Lambda \epsilon^{ij} \mathcal{E}_j)$, where $\mathcal{E}_i = 0$ is the equation of motion for the nonlinear shallow water (8). Hence, the Noether current is conserved on-shell. By

---

[7]We note that from a condensed matter perspective, the sharp boundary conditions yielding (28) can be delicate, e.g. see [23].

virtue of the EoM and as in any gauge theory, the Noether current can be written as a total derivative, $J^\mu_\Lambda \approx \partial_\nu \Gamma^{\mu\nu}_\Lambda$,

$$\Gamma^{0i}_\Lambda = \Lambda \epsilon^{ij} u_j\,, \qquad \Gamma^{ij}_\Lambda = -\epsilon^{ij}\left(\frac{u^2}{2} + gh\right)\Lambda\,. \tag{34}$$

The "local circulation", the charge associated with the above Noether current for a generic $\Lambda$, is then given by

$$\Gamma_\Lambda = \int_\Sigma d\Sigma_\mu J^\mu_\Lambda = \int_\Sigma d\Sigma_\mu \partial_\nu \Gamma^{\mu\nu}_\Lambda = \oint_\mathcal{B} d\Sigma_{\mu\nu} \Gamma^{\mu\nu}_\Lambda\,. \tag{35}$$

By explicit form of $\Gamma^{\mu\nu}_\Lambda$ in hand (34), one can compute the surface charge (35),

$$\boxed{\Gamma_\Lambda = \oint_\mathcal{B} \Lambda\, u \cdot dl\,.} \tag{36}$$

For constant $\Lambda$, $\Gamma_\Lambda$ reduces to (3).[8] To proceed, we assume that the surface $\Sigma$ has a disk topology and its boundary $\mathcal{B}$ is parametrized by $\phi \in [0, 2\pi]$. We define the *circulation aspect charge* $\gamma$ as follows

$$\Gamma_\Lambda = \oint_\mathcal{B} \Lambda\gamma\, d\phi\,, \qquad \gamma\, d\phi := u \cdot dl\,. \tag{37}$$

To connect $\Gamma_\Lambda$ to the circulation $\Gamma_\mathcal{C}$ (4), we note $\Lambda$ is an arbitrary (time-independent) function and hence for any given $\mathcal{C}$ there exists a $\Lambda$ such that $\Gamma_\Lambda = \Gamma_\mathcal{C}$. Therefore, the constancy of $\Gamma_\Lambda$ provides us with a reinterpretation of the Kelvin circulation theorem in terms of Noether's theorem and conserved charges.

As is well known in gauge theories, e.g. see [24, 25], not all gauge field configurations related by gauge transformations are physically equivalent. As a nomenclature, the gauge fields related by proper or trivial gauge transformations, those leading to zero surface charges, are physically equivalent, whereas those related by physical or improper gauge transformations, that lead to non-zero surface charges, are physically inequivalent. In this sense, and in our example, gauge transformations $\Lambda$ appearing in (37) are nontrivial gauge transformations. Apparently, one can use the surface charge $\Gamma_\Lambda$ to label gauge field configurations which differ by a gauge transformation.

**Charge algebra.** One can read the algebra of the surface charge (36) by using the standard definition of the Poisson bracket

$$\{\Gamma_{\Lambda_1}, \Gamma_{\Lambda_2}\} = \delta_{\Lambda_2}\Gamma_{\Lambda_1}\,. \tag{38}$$

From the fact that the charge aspect $\gamma$ (37) is gauge invariant, as expected, we obtain a $u(1)$ algebra

$$\{\gamma(t, \phi), \gamma(t, \phi')\} = 0\,. \tag{39}$$

## 3.2 Linear theory

Symmetries and edge modes for the Maxwell-Chern-Simon theory have been studied, see e.g. [26–28] and especially [29]. Here we focus on the surface charges of the theory and

---

[8]Note that the $f$ term in (3) may also be added to the above for time-independent gauge parameter $\Lambda$. That is, one may add $-f\frac{\Lambda}{2}\epsilon^{ij}x_j$ with $\partial_t\Lambda = 0$ into the integral and still get a conserved charge: $\Gamma_\Lambda = \oint_\mathcal{B} \Lambda(u^i - \frac{f}{2}\epsilon^{ij}x_j)\,dl_i$.

construct the Noether charge associated with gauge symmetries for the MCS theory. The analysis is similar to what we have done for the nonlinear shallow water system in the previous subsection.

The starting point is the Noether current associated with the gauge symmetry

$$J^\mu_\Lambda = -g\left(F^{\mu\nu} - \frac{f}{2v}\epsilon^{\mu\nu\rho}A_\rho\right)\partial_\nu\Lambda - \frac{vf}{2H}\Lambda\epsilon^{\mu\nu\rho}\partial_\nu A_\rho\,. \tag{40}$$

The divergence of this Noether current is proportional to the EoM (16), $\partial_\mu J^\mu_\Lambda = -\mathrm{E}^\mu\partial_\mu\Lambda$ and hence is conserved on-shell. By applying the EoM, we get the following result

$$J^\mu_\Lambda \approx \partial_\nu\Gamma^{\mu\nu}_\Lambda\,, \qquad \Gamma^{\mu\nu}_\Lambda := -g\Lambda\left(F^{\mu\nu} - \frac{f}{2v}\epsilon^{\mu\nu\rho}A_\rho\right)\,. \tag{41}$$

We should crucially note that the above may not be obtained from (34) in the linearized regime. This may be understood by noting that to obtain the linearized MCS theory (14) we are eliminating $\tilde{A}_\mu$ gauge field in favor of $A_\mu$ and that in the nonlinear theory there are two $u(1)$ currents. Eq.(34) is one of them and we did not discuss the other $u(1)$ current. Our linearization, which involves also the elimination of $\tilde{A}_\mu$, mixes up the two $u(1)$ currents which yield the $f/v$ term in (41).

Finally, the Noether charge associated with gauge transformation $\Lambda$ is expressed as a co-dimension two integral

$$\Gamma_\Lambda = -g\oint_\mathcal{B} d\Sigma_{\mu\nu}\Lambda\left(F^{\mu\nu} - \frac{f}{2v}\epsilon^{\mu\nu\rho}A_\rho\right)\,. \tag{42}$$

As the gauge parameter $\Lambda$ is an arbitrary function, we can interpret this surface charge as an infinite number of charges for the MCS gauge theory. By using the dictionary (12) for $Q = 0$ we can find the expression of the surface charges in terms of the Poincare wave variables

$$\boxed{\Gamma_\Lambda = \oint_\mathcal{B} \Lambda\left(u - \frac{f}{2H}A\right)\cdot dl\,.} \tag{43}$$

The surface charge (43) is not gauge invariant, to see this explicitly, we note that

$$\Gamma_\Lambda = \oint_\mathcal{B} \Lambda\gamma\, d\phi\,, \qquad \gamma\, d\phi := \left(u - \frac{f}{2H}A\right)\cdot dl = -\frac{1}{H}\left(\epsilon_{ij}E_j + \frac{f}{2}A_i\right)dl_i\,, \tag{44}$$

where we used (12). This charge aspect changes under gauge transformation $\Lambda$ as follows

$$\delta_\Lambda\gamma = -\frac{f}{2H}\partial_\phi\Lambda\,. \tag{45}$$

This gauge non-invariance of the charge aspect is a familiar feature of Chern-Simons theories and as we show below it yields a central extension term in the charge algebra.

**Charge algebra.** For the MCS theory the charge algebra is more interesting as it takes a central extension term. To get the charge algebra, we use the standard Poisson bracket (38). Recalling the surface charge expression (43), we find the following result

$$\boxed{\{\gamma(t,\phi),\gamma(t,\phi')\} = -\frac{f}{2H}\partial_\phi\delta(\phi - \phi')\,.} \tag{46}$$

This is the standard Kac-Moody algebra at a level proportional to $f/H$. ]We remark that the appearance of the Kac-Moody level is a result of the elimination of the $\tilde{A}_\mu$ gauge field and its gauge symmetry, cf. discussions below (41).

**Global charge.** The circulation charge is the global $\Lambda = 1$ element in the charge $\Gamma_\Lambda$

$$\Gamma = \oint_{\mathcal{B}} d\phi \left( u_\phi - \frac{f}{2H} A_\phi \right), \tag{47}$$

reproducing (3). Under gauge transformation $\Lambda$, this global charge transforms as

$$\delta_\Lambda \Gamma = -\frac{f}{2H} \oint_{\mathcal{B}} d\phi \,, \qquad \partial_\phi \Lambda = -\frac{f}{2H} \Delta\Lambda \,. \tag{48}$$

Therefore, for the single-valued gauge transformations, $\Delta\Lambda = 0$, the global charge (circulation) is gauge invariant, $\delta_\Lambda \Gamma = 0$. That is, the zero mode of the charge is a central element in the Kac-Moody charge algebra.

# 4 Lagrange description of linearized shallow water

As discussed in the opening of section 2, the gauge theoretic formulation provides us with a dual description for the fluid system, in the sense that the two Bianchi identities for the $u(1) \times u(1)$ gauge fields appear as the Noether conservation relations of the fluid system. In this section, we take a different viewpoint and try to find a different physical meaning for the origin of the $u(1)$ gauge symmetry in the shallow water system. This viewpoint becomes apparent in the Lagrange (comoving) description of the fluid, while so far we have worked in the Euler description. As we discuss below, the symmetry of relabelling of fluid particles [30–32] gives rise to a part of $u(1)$ symmetry one remains with after fixing the temporal gauge [33, 34].

To find the Lagrange description of linearized shallow water, we follow ideas outlined in [33, 34] and start with

$$u_i = -\frac{1}{H} \epsilon_{ij} E_j \,. \tag{49}$$

In the temporal gauge, we have

$$u_i = -\frac{1}{H} \epsilon_{ij} \partial_t A_j \,. \tag{50}$$

Next, we introduce the comoving coordinates $y^i$:

$$u_i(x(y,t),t) = \frac{\partial}{\partial t} x_i(y,t), \tag{51}$$

where $y$ stands for the label of particles constituting the fluid (it could be the initial position of particles). We can hence write (50) as

$$\frac{\partial}{\partial t} x_i(y,t) = -\frac{1}{H} \epsilon^{ij} \left. \frac{\partial A_j(x,t)}{\partial t} \right|_{x=x(y,t)} \,. \tag{52}$$

Integrating over time we obtain,

$$x^i(y,t) - x^i(y,0) = -\frac{\epsilon^{ij}}{H} \int_0^t dt \left. \frac{\partial A_j(x,t)}{\partial t} \right|_{x=x(y,t)} \,. \tag{53}$$

Here we assume the change of the gauge field (or velocity field) on the probe's path is slowly varying. In the leading order, we can approximate the integrand of the RHS as $\left. \frac{\partial A_j(x,t)}{\partial t} \right|_{x=x(y,t)} = \frac{\partial A_j(y,t)}{\partial t}$ (see section 7 for more discussion), and finally by performing the time integral, we get

$$x^i(y,t) - x^i(y,0) = -\frac{\epsilon^{ij}}{H}\left[A_j(y,t) - A_j(y,0)\right].\tag{54}$$

In the Lagrange description, $x^i = y^i$ may be chosen as the equilibrium state of the fluid. This implies, $x_i(y,0) = y_i$, $A_j(y,0) = 0$. We then find

$$x^i(y,t) = y^i - \frac{1}{H}\epsilon^{ij}A_j(y,t).\tag{55}$$

The above is written in a specific (temporal) gauge. Since the initial point (50) is a linear order equation, the relation among $x, y$ coordinates and the gauge field $A$ works at the linear level. In the Lagrange description we are dealing with fields of comoving coordinates $y$ while in the Euler description of previous sections with a field theory over the $x$ space.

Introduction of the comoving coordinates $y$ provides a geometric meaning to the "residual gauge transformations" on $A_i$ (that one remains with after fixing the temporal gauge) in the Lagrange description: gauge transformations on $A_i(y,t)$ can be equivalently viewed as a class of 2 dimensional $y$-space diffeomorphisms [34],

$$y^i \to y^i + \xi^i, \qquad \xi^i = -\frac{1}{H}\epsilon^{ij}\partial_j\Lambda,\tag{56}$$

where $\partial_i = \frac{\partial}{\partial y^i}$ and $\Lambda = \Lambda(y^i)$. It is manifest that $\partial_i\xi^i = 0$. Therefore, we are dealing with area-preserving diffeomorphisms (APDs). Note that in the 2d sense, a scalar ($u(1)$ gauge transformations $\Lambda$) is Hodge-dual to a divergence-free vector (2d APDs). One should note that this "residual" symmetry arises as the continuum limit of the freedom in the labeling of fluid particles in the Lagrange description. As discussed, one can associate conserved charges to these residual symmetries which can label physical states. The APD is therefore equivalent to the improper part of $u(1)$ gauge symmetry of the MCS theory in Lagrange description of the linearized fluid.

# 5 Fluid memory effect for linearized theory

To explore the physical and possibly observational meaning of the infinite set of surface charges we discussed in the previous section, as in other similar cases in gauge and gravity theories, we associate the variations in these charges to memory effect in the fluid system. In this section we study such a memory in two ways: (1) Memory from EoM; (2) Memory from conservation. This analysis will be completed and extended in the next two sections.

## 5.1 Memory, fluid waves and relevant scales

Memory is a "permanent" (long time scales) trace remaining in a system after a perturbation or fluctuation has passed. When we deal with a massless theory, like Maxwell or Einstein gravity, with fluctuations being various photon or graviton wave-packets, the memory is in low frequency end of the spectrum in the wave-packet. If we denote the time scale of the measurement by $T$, theoretically $T \to \infty$, frequencies relevant to the memory effect are IR modes with $\omega T \sim 1$. For the massless waves with dispersion relation $\omega = vk$ low frequency means very long wave-length $\lambda$ such that $\lambda \sim vT$.

In the linearized shallow water system, the waves do not typically have a linear dispersion relation and are propagating in the fluid which has a finite size. Let us then review the time and space scales relevant to the memory effect in this system. Consider a body of fluid with depth/height $H$ and 2 dimensional span $L^2$. Shallow water approximation then requires

$L \gg H$. The inverse of the Coriolis parameter $f$ defines a time scale in the system, which for oceans on the Earth is a day or more (depending on the altitude, see footnote 1). This time scale then yields the other relevant length scale of the system, the Rossby radius of deformation $R = v/f$ [21]. It denotes the length the phase of a wave can travel in a day or the length scale where the effect of Coriolis force and gravity wave (or buoyancy) are comparable.[9] For typical shallow water systems in linearized approximation, $R \gtrsim L \gg H$.

Besides the characteristic length and time scales of the system, there are length and time scales associated with the waves or "fluctuations" in the system, the frequency $\omega$, and the wavelength $\lambda$. In the shallow water system, there are in particular the Kelvin coastal waves and the Poincare waves, reviewed in section 2.3. The former describes waves propagating along a coast (one dimensional waves) with dispersion relation $\omega = vk$ and the latter, in the linearized theory, are described by two dimensional "massive modes" with dispersion relation $\omega^2 = v^2 k^2 + f^2$. There is hence a low-frequency cutoff $f$ for the Poincare waves. Note that for the Poincare waves the energy of the wave of a given amplitude $h$ and wave-length $\lambda$ is proportional to $1 + 2\lambda^2/R^2$, cf. (25).[10] So, the notions of low energy and long wave-length are not synonymous. Unlike the massless case (and Kelvin coastal waves in our case), we do not have the "soft" modes with arbitrarily small frequencies, theoretically $\omega \to 0$.

More importantly, unlike the usually studied memory effects for gravity and electromagnetism, wave-packets in a shallow water system can't have arbitrarily long wave-lengths, they are subject to the size of the system, $\lambda \leq 2L$. This introduces a low-frequency cutoff, $\omega > v/\lambda > v/(2L)$. Moreover, as discussed, for Poincare waves $f$ appears as another low-frequency cutoff $f$, $\omega > v/R$. So, the lowest frequency mode is given by $\min(v/R, v/(2L))$ and for the typical cases where $R \gtrsim L$, it is associated with the longest wave-length $2L$. On the other hand, to be able to detect traces of such modes in a time $T$ we should have $\omega T \sim 1$. To summarize, the memory effect can be detectable if $f T \gtrsim 1$ and $\lambda \sim L \sim R$. One should, however, note that typical modes in the system have $\lambda \lesssim L \lesssim R$.[11]

## 5.2 Memory from EoM

The linearized equation of motion (9) at vanishing vorticity $Q = 0$ takes the form[12]

$$\partial_t u_i = f \epsilon_{ij} u^j - \frac{gH}{f} \epsilon^{kl} \partial_i \partial_k u_l. \tag{57}$$

Next recall that $u^i = -\frac{1}{H} \epsilon^{ij} E_j$, which in the temporal gauge $A_t = 0$ takes the form $u_i = -\frac{1}{H} \epsilon^{ij} \partial_t A_j$. Note that in the temporal gauge we are left with time-independent residual gauge transformations $\Lambda = \Lambda(x)$. The above equation in terms of the gauge field $A_i$ and the associated circulation charge aspect $\gamma_i = u_i - \frac{f}{2H} A_i$ can be written as,

$$\partial_t \gamma_i = \frac{f}{2H} \partial_t A_i - \frac{g}{f} \partial_i \partial_j \partial_t A_j. \tag{58}$$

One can immediately integrate (58) to get

$$\Delta \gamma_i = \frac{g}{f} \left( \frac{1}{\ell^2} \Delta A_i - \partial_i (\nabla \cdot \Delta A) \right), \tag{59}$$

---

[9] For the oceans on the Earth with depth $H = 1km$ at around altitude $\theta = 45°$, $v \simeq 100m/sec$, $f \simeq 10^{-5}/sec$ and $R = 10^4 km$ which is of the order of Earth radius but much larger than the depth of the ocean.

[10] Note also that for typical waves, $\lambda \ll R$, and the energy density in the wave is essentially independent of the wave-length. As discussed in section 2.3 the same is true for Kelvin coastal waves.

[11] If we have a wave-packet, the spatial (or temporal) width of the wave-packet also introduces another scale in the system, which we typically take them to be much smaller than $L$ (or $T$).

[12] Had we considered $Q \neq 0$ sector, one should have added $\frac{g}{f} \partial_i Q$ term to the right-hand-side of (57).

where $\ell := \sqrt{2}R$ with $R$ being the Rossby radius and $\Delta A_i$ is the *memory field*,

$$\Delta A_i := A_i(+T) - A_i(-T), \tag{60}$$

with $+T$ and $-T$ denoting the late and early times and we used $gH = v^2$. Using the Green's function analysis in the appendix A one may invert (59) and solve the memory field for $\Delta\gamma_i$,

$$\Delta A_i(x) = \frac{f}{g} \int d^2x' \, G_{ij}(x - x')\Delta\gamma_j(x'). \tag{61}$$

## 5.3 Memory from conservation

Here we work out the relation between memory fields and conservation of the fluid surface charges (43) in the Maxwell-Chern-Simon theory. We start with the conservation of the surface charge $\Gamma_\Lambda$ (43)

$$\Delta\Gamma_\Lambda = \oint_{\mathcal{B}} \Lambda\Delta\gamma \cdot dl, \tag{62}$$

where $\Delta\Gamma_\Lambda = \Gamma_\Lambda(+T) - \Gamma_\Lambda(-T)$ and we have assumed $\partial_t\Lambda = 0$. From the EoM in terms of the memory field (59), we obtain the conservation-memory equation

$$\Delta\Gamma_\Lambda = \frac{f}{2H} \oint_{\mathcal{B}} \Lambda\big[\Delta A - \ell^2\nabla(\nabla \cdot \Delta A)\big] \cdot dl, \tag{63}$$

which relates to the permanent change in the fluid charges due to the passage of fluid waves parameterized through the memory field $\Delta A$. It is one of the main results of this paper. The tangent component of (59) on $\mathcal{B}$ may be recovered from (63) for $\Lambda = \delta(\phi - \phi')$. We note that $\ell$ is much larger than the typical wave-length of the waves and as such the second term in (63) dominates over the first term and $\Delta\Gamma_\Lambda \simeq -\frac{f}{g}\oint_{\mathcal{B}} \Lambda\nabla(\nabla \cdot \Delta A) \cdot dl$. Nonetheless, for the wave lengths relevant to the memory effect, cf. discussion in section 5.1, we are dealing with waves in which these two terms are of the same order.

## 6 Memory from probes, memory at nonlinear level and Stokes drift

Lagrange description of a fluid, cf. section 4, is the convenient framework to follow path-line of probe particles and the related memory effect. Consider a probe particle in the fluid with the trajectory $x_i(y,t)$ (where $y$ denotes the initial position of the probe). At any time $t$, the velocity of this particle is given by the velocity field $u$ evaluated at the position of the particle, i.e.

$$\frac{dx^i(y,t)}{dt} = u^i(x(y,t),t) = -\frac{1}{H}\epsilon^{ij}E_j(x(y,t),t) = -\frac{1}{H}\epsilon^{ij}\frac{\partial A_j(x,t)}{\partial t}\bigg|_{x=x(y,t)}. \tag{64}$$

where we used (12) and in the last equality we fixed the temporal gauge.

We can now solve this equation perturbatively. At the second order of perturbation the path-line equation yields

$$\Delta x^i = -\frac{\epsilon^{ij}}{H}\Delta A_j + \frac{\epsilon^{jk}\epsilon^{il}}{H^2}\int_{-T}^{+T} dt \, \partial_j E^l(y,t)\Delta A_k(t) + \mathcal{O}(u^3), \tag{65}$$

where $\Delta x^i = x^i(y, T) - x^i(y, -T)$, $\Delta A_i(t) = A_i(y, t) - A_i(y, -T)$ and $\Delta A_i = \Delta A_i(T) = A_i(y, T) - A_i(y, -T)$. It shows how one may probe memory field (60) by studying the displacement of particles in the fluid. We will show next that this displacement which we dub as *path-line memory*, is related to the Stokes drift.

## 6.1 Stokes drift as fluid memory

Assume the velocity and its time variations are small. To systematically explore this we consider

$$\frac{\mathrm{d}x^i(t)}{\mathrm{d}t} = \epsilon\, u^i(x(t), t), \tag{66}$$

and make an expansion in powers of $\epsilon$. One can perturbatively solve this equation for $x_i(t)$,

$$x^i(t) = \sum_{a=0}^{\infty} \epsilon^a x_a^i(t) = x_0^i(t) + \epsilon x_1^i(t) + \epsilon^2 x_2^i(t) + \dots \tag{67}$$

By substituting this perturbative expansion in (66), one obtains the following result in the zeroth order of expansion

$$\frac{\mathrm{d}x_0^i(t)}{\mathrm{d}t} = 0 \quad \Longrightarrow \quad x_0^i(t) = y^i. \tag{68}$$

In the first order we find,

$$\frac{\mathrm{d}x_1^i(t)}{\mathrm{d}t} = u^i(y, t) \quad \Longrightarrow \quad x_1^i(t) = \int_0^t \mathrm{d}t_1 u^i(y, t_1), \tag{69}$$

and the second order yields,

$$\frac{\mathrm{d}x_2^i(t)}{\mathrm{d}t} = x_1^j(t)\partial_j u^i(y, t) \quad \Longrightarrow \quad x_2^i(t) = \int_0^t \mathrm{d}t_1\, \partial_j u^i(y, t_1) \int_0^{t_1} \mathrm{d}t_2\, u^j(y, t_2). \tag{70}$$

Finally, we have

$$\begin{aligned} \Delta x^i = x^i - x_0^i &= \sum_{a=1}^{\infty} \epsilon^a x_a^i(t) \\ &= \int_{-T}^{+T} \mathrm{d}t\, u^i(y, t) + \int_{-T}^{+T} \mathrm{d}t \int_{-T}^{t} \mathrm{d}t'\, u(y, t') \cdot \nabla u^i(y, t) + \mathcal{O}(u^3). \end{aligned} \tag{71}$$

This equation, once written in terms of the gauge fields, as expected, is the same as (65).

Let us now explore the physical meaning of the two terms appearing in (65). To this end, recall the definition of the Stokes drift $\Delta x_S$ (e.g. see [15, 35])

$$\Delta x_S := \Delta x_L - \Delta x_E, \tag{72}$$

where $\Delta x_L$ and $\Delta x_E$ are respectively Lagrange and Euler displacements (drifts). $\Delta x_L$ is the left-hand-side of (65),

$$\Delta x_E^i = \int_{-T}^{+T} \mathrm{d}t\, u^i(y, t) = -\frac{\epsilon^{ij}}{H}\Delta A_j, \tag{73}$$

and

$$\begin{aligned} \Delta x_S^i &= \int_{-T}^{+T} \mathrm{d}t\, \Delta x_E^j(y, t) \cdot \partial_j u^i(y, t) \\ &= \frac{1}{H^2}\left(\frac{1}{2}\nabla_i(\Delta A)^2 + \epsilon^{ij}\Delta A_j B(y, -T) - \partial_j \int_{-T}^{+T} \mathrm{d}t\, E_j \Delta A_i(t) - \epsilon^{ij} \int_{-T}^{+T} \mathrm{d}t\, E^j B\right), \end{aligned} \tag{74}$$

where $E_i, B = \epsilon^{jk}\partial_j A_k, \Delta A_i$ which appear under the integrals are functions of $t, y$ and $\Delta A_i$ which appears outside the integral is only a function of $y$. Each of the four terms in the above are explicitly invariant under time-independent gauge transformations $A_i \to A_i + \partial_i \Lambda, \Lambda = \Lambda(y)$.

The Stokes drift $\Delta x_S$ consists of two parts: A part which does not involve time integral and the part which involves time integrals. Following usual terminology of the memory effect, let's dub the former the "soft part" and the latter as the "hard part". The soft part has two terms. Assuming that the passing wave has a short (finite) time span, $B$ and $E_i$ of the passing wave vanishes at $-T$ or $T$. Therefore, one may drop $\Delta AB(y, -T)$ term and the soft part of the Stokes drift is given by $\frac{1}{2H^2}\nabla_i(\Delta A)^2$. The hard part, consists of two terms in which $\int \epsilon^{ij}E_j B$ term is the Pointing vector for the passing wave. This term in the Stokes drift measures how much energy is transferred to the system due to the passage of the wave.

## 6.2 Coastal Kelvin wave, Euler and Stokes drifts

One may now compute the Euler and Stokes drifts (74) for the Kelvin wave configuration (31) and (32). The Euler displacement is given by

$$\Delta x_E^1 = 0, \qquad \Delta x_E^2 = e^{-\frac{x_1}{R}} \int_{-T}^{+T} \mathrm{d}s \, U(vs). \tag{75}$$

Assuming that $U(x_2 + vt)$ vanishes at far past, and noting that $E_i, A_i$ have only $x_1$-component, it is readily seen that the first three terms in the second line of (74) vanish and only the Pointing vector term contributes:

$$\Delta x_S^1 = 0, \qquad \Delta x_S^2 = -\frac{e^{-\frac{2x_1}{R}}}{v} \int_{-T}^{+T} \mathrm{d}s \, U^2(vs) = -\frac{1}{v}\int_{-T}^{+T} \mathrm{d}s \, u_2^2. \tag{76}$$

The above results are worked out in the linear theory. One may perform the above computation in a non-perturbative way. To this end, we start from

$$\frac{\mathrm{d}x_2}{\mathrm{d}t} = e^{-\frac{x_1}{R}} U(x_2 + vt).$$

If we call $s = t + x_2/v$ and replace $t$ for $s$ and assume $T \gg x_2/v$ we learn,[13]

$$\begin{aligned}
\Delta x_2 &= e^{-\frac{x_1}{R}} \int_{-T}^{+T} \mathrm{d}s \, \frac{U(s)}{1 + \frac{1}{v}e^{-\frac{x_1}{R}}U(s)} \\
&= e^{-\frac{x_1}{R}} \int_{-T}^{+T} \mathrm{d}s \, U(s)\left(1 - \frac{U(s)}{v}e^{-\frac{x_1}{R}} + \mathcal{O}(\frac{U^2}{v^2})\right) = \Delta x_E^2 + \Delta x_S^2 + \mathcal{O}\left(\frac{U^3}{v^3}\right).
\end{aligned} \tag{77}$$

As an explicit example, consider the Gaussian wave-packet,

$$U(vs) = u_0 e^{-\frac{v^2 s^2}{b^2}}, \tag{78}$$

where $u_0$ and $b$ are constants with the dimensions of velocity and length respectively. With this choice, we find the following result for the Euler and Stokes drifts (taking $T \to \infty$ limit),

$$\Delta x_E^2 = \sqrt{\pi}\frac{u_0}{v}b e^{-\frac{x_1}{R}}, \qquad \Delta x_S^2 = -\sqrt{\frac{\pi}{2}}\frac{u_0^2}{v^2}b e^{-\frac{2x_1}{R}}. \tag{79}$$

As we see Euler and Stokes drifts are respectively first and second order in $u_0/v$.

---

[13]As mentioned, Kelvin coastal waves are essentially $1+1$ dimensional phenomena and may hence be studied in $1+1$ dimensional fluids. See [36] for analysis of the Stokes drift in a particular $1+1$ dimensional fluid.

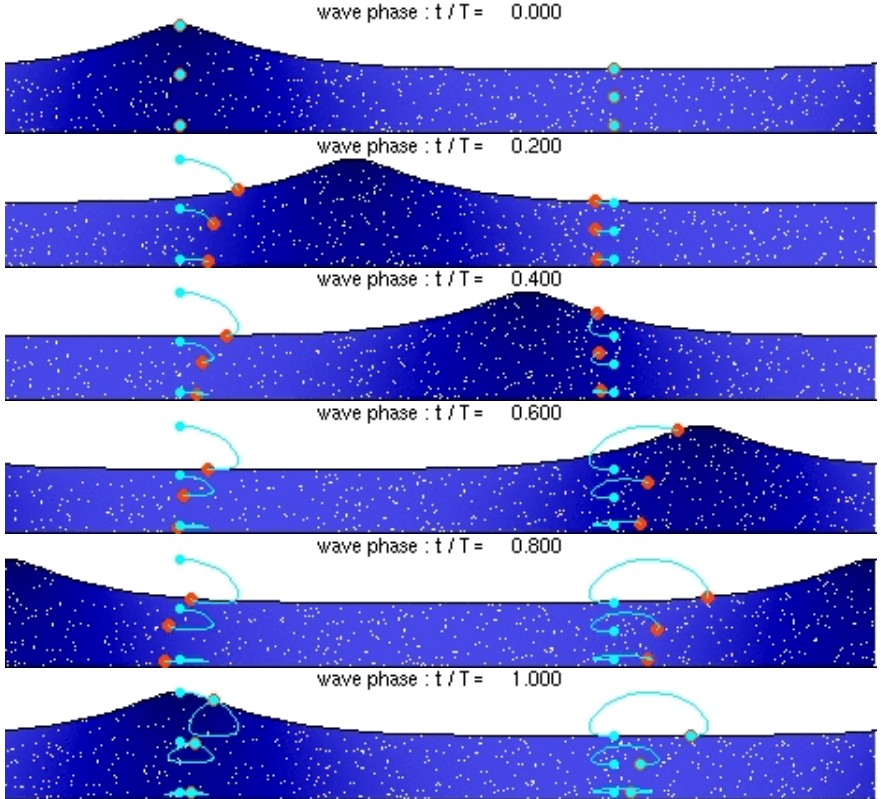

Figure 2: Stokes drift for a passing wave in one period. Courtesy of [37].

## 6.3 Poincare waves, Euler and Stokes drifts

Consider the case depicted in Fig. 2 and take $\Delta T = 2\pi/\omega$. The Eulerian displacement due to the passage of a Poincare wave, recalling (73) and (23), is given by

$$\Delta x^i_E = -\frac{1}{H}\epsilon^{ij}\Delta A_j = -\frac{1}{H}\epsilon^{ij}\left(A_j(x,T) - A_j(x,0)\right) = 0. \tag{80}$$

One may compute the Stokes drift using (74). Since the memory field $\Delta A_i$ vanishes over a period, the "soft part" of the drift, the first two terms in (74), vanish. One can also show that the third term in (74) also yields a zero integral and the drift is only receiving contribution from the fourth term, the Pointing vector term:

$$\Delta x^i_s = -\frac{1}{H^2}\epsilon^{ij}\int_0^{2\pi/\omega} \mathrm{d}t\; E_j B = \frac{\pi h^2}{H^2 k^2}k_i. \tag{81}$$

As expected, the drift is along the direction of the wave. Its magnitude is proportional to the square of the ratio of the amplitude of the waves to the height of water and is linearly proportional to its wave-length, so the drift is larger for longer wave-lengths. Recalling (25) and that the energy density transferred through the Poincare waves is $\Delta\mathcal{E} = \frac{h^2}{H^2}\frac{v^2}{2}\frac{2}{k^2 R^2} = \frac{h^2}{H^2}\frac{f^2}{k^2}$, then $\Delta x^i_s = \pi\Delta\mathcal{E}k_i/f^2$.

One may consider a smooth wave-packet with a short time span compared to the total time $T$. Since the Poincare waves have frequencies bigger than $f$, $\omega \geq f$, the memory field $\Delta A_i$ for a generic wave packet will be zero and therefore, Euler displacement $\Delta x^i_E \simeq 0$. For a similar reason, the Stokes drift (74) is given by the Pointing vector (last term) and is proportional to the energy transferred by the wave and is along the wave direction.

# 7 Memory in forced linearized shallow water, Darwin drift

In shallow water systems Darwin drift[14] [16] refers to the permanent displacement of a fluid parcel as a result of the passage of a body through a fluid. See the animation in [38]. To study this, we need to consider the shallow water system in the presence of an external force $\mathcal{F}_i$ density which may be formulated by,

$$\partial_t \eta + H \nabla \cdot u = 0, \tag{82a}$$

$$\partial_t u_i = f \epsilon_{ij} u^j - g \partial_i \eta + \mathcal{F}_i. \tag{82b}$$

Kelvin's circulation theorem implies that $\mathcal{F}_i$ should come from gradient of a potential density, pressure field $P$, i.e. $\mathcal{F}_i = -\partial_i P$. These equations may be obtained from the action

$$S = -\frac{g}{4\nu} \int d^3x \left( F^{\mu\nu} F_{\mu\nu} - \frac{f}{\nu} \epsilon^{\mu\nu\rho} A_\mu F_{\nu\rho} + \frac{2}{g} \epsilon^{ij} F_{ij} P \right), \tag{83}$$

which is a direct generalization of (14). The gauge invariance of the above action is a manifestation of Kelvin's circulation theorem.

With this gauge-invariant action, we can compute the relation between the memory and EoM,

$$\Delta \gamma_i = \frac{g}{f} \left( -\partial_i (\nabla \cdot \Delta A) + \frac{1}{\ell^2} \Delta A_i \right) - \partial_i \Delta P, \qquad \Delta P(x) := \int_{-T}^{+T} dt\, P(x, t), \tag{84}$$

and

$$\Delta u_i = \frac{g}{f} \left( -\partial_i (\nabla \cdot \Delta A) + \frac{1}{R^2} \Delta A_i \right) - \partial_i \Delta P. \tag{85}$$

This equation can be solved for $\Delta A_i$ as follows

$$\begin{aligned}
\Delta A_i(x) &= -\frac{f}{g} \int d^2x' \, \tilde{G}_{ij}(x - x') \left[ \Delta u_j(x') + \partial_j \Delta P(x') \right] \\
&= \frac{1}{f} \epsilon_{ij} \Delta E_j(x) - \frac{H}{f} \partial_i \mathcal{X}(x),
\end{aligned} \tag{86}$$

where $\tilde{G}_{ij}(X) = \frac{R^2}{(2\pi)^2} \partial_i \partial_j I(X) + R^2 \, \delta_{ij} \delta^2(X_i)$ with $I(X) = 2\pi K_0(X/R)$, see appendix A for more details, $\Delta E_i = E_i(x, T) - E_i(x, -T)$ and

$$\mathcal{X} = \Delta P(x) + \frac{1}{2\pi} \partial_i \int d^2x' \, K_0\left( \frac{|x - x'|}{R} \right) \left( -\frac{1}{H} \epsilon^{ij} \Delta E_j(x') + \partial_i \Delta P(x') \right). \tag{87}$$

Eq.(86) indicates that if $\Delta E_i$ vanishes, then $\Delta A_i$ is a pure gauge transformation.

As the next task, we relate the memory field to the conservation of the circulation charge aspect

$$\Delta \Gamma_\Lambda = \frac{f}{2H} \oint_\mathcal{B} \Lambda \left[ \Delta A - \ell^2 \nabla(\nabla \cdot \Delta A) - \frac{2H}{f} \nabla(\Delta P(x')) \right] \cdot dl, \tag{88}$$

which is a direct generalization of (63) for the forced case. Finally, we connect the displacement memory effect to the change of charge aspect in presence of the external potential,

$$\begin{aligned}
\Delta x^i &= \frac{f}{\nu^2} \epsilon^{ij} \int d^2y \, G_{jk}(x - x') \left[ \Delta \gamma_k(x') + \partial_k \Delta P(x') \right] \\
&= \frac{2}{f} \epsilon^{ij} \left[ \Delta \gamma_j(x) + \partial_j \Delta P(x) + \frac{1}{2\pi} \partial_j \int d^2x' \, K_0\left( \frac{|x - x'|}{\ell} \right) \partial_k \left( \Delta \gamma_k(x') + \partial_k \Delta P(x') \right) \right].
\end{aligned} \tag{89}$$

---

[14]This was introduced in 1953 by Charles G. Darwin who is a grandson of the renowned biologist Charles R. Darwin.

We note that this is a first order displacement (unlike the Stokes drift which is second order).[15] The terms proportional to $\Delta P$ give the Darwin drift [16,17]:

$$
\begin{aligned}
\Delta x^i_{{}_D} &= \frac{2}{f}\epsilon^{ij}\partial_j\left[\Delta P(x)+\frac{1}{2\pi}\nabla^2\int \mathrm{d}^2x'\, K_0\left(\frac{|x-x'|}{\ell}\right)\Delta P(x')\right]\\
&= \frac{1}{2\pi}\frac{f}{gH}\epsilon^{ij}\partial_j\int \mathrm{d}^2x'\, K_0\left(\frac{|x-x'|}{\ell}\right)\Delta P(x').
\end{aligned}
\tag{90}
$$

**Memory implants in shallow water.**  Consider an external potential with a shock-wave profile

$$
P = \bar{P}(x)\delta(t-t_0),
\tag{91}
$$

so that $\Delta P(x)=\bar{P}(x)$. Next, assume that the velocity field is zero at the initial times $(-T\ll t_0)$ long before the wave turns on and also at the late times $(+T\gg t_0)$ when the wave effect has left the system, i.e. $\Delta u_i = 0$. Then, (85) yields

$$
\frac{f}{H}\Delta A_i - \frac{g}{f}\partial_i(\nabla\cdot\Delta A)-\partial_i\bar{P}=0.
\tag{92}
$$

$\Delta u_i = 0$ also implies $\Delta E_i = 0$, that is, the initial and final states of the fluid can differ only up to gauge transformations, $\Delta A_i = \partial_i\Lambda$. $\Lambda$, however, is not an arbitrary function and depends on the external potential pressure. From (86) one learns,[16]

$$
\begin{aligned}
\Lambda(x) &= -\frac{H}{f}\left[\bar{P}(x)+\frac{1}{2\pi}\nabla^2\int \mathrm{d}^2x'\, K_0\left(\frac{|x-x'|}{R}\right)\bar{P}(x')\right]\\
&= -\frac{f}{2\pi g}\int \mathrm{d}^2x'\, K_0\left(\frac{|x-x'|}{R}\right)\bar{P}(x').
\end{aligned}
\tag{93}
$$

The above shows how the shallow water system responds to a shock-wave, where the initial and final states are both stationary and have the same velocity. These states differ by a large gauge transformation (93) which arises due to the external potential.[17] In other words, these equations describe how the shallow water remembers the traces of the passage of the shock. In this regard, we call these equations the *memory implant equations*. We finally note that (90) and (93) are related exactly as given in (56). That is, the Darwin drift may be understood as a area-preserving diffeomorphism in the Lagrange description of the fluid.

We end by showcasing computation of the Darwin drift (90) for functions $\bar{P}(x)$. Let $\bar{P}(k)$ denote the Fourier transform of $\bar{P}(x)$ and assume $\bar{P}(k)$ is only a function of $|k|$, then

$$
\Delta x^i_{{}_D} = \frac{2\pi f}{gH}\epsilon^{ij}\partial_j\int \mathrm{d}k\,\frac{k}{k^2+1/\ell^2}J_0(kr)\bar{P}(k),
\tag{94}
$$

where $r=|x|$. The above may be written as

$$
\Delta x^\phi_{{}_D} = \frac{f}{gH}\frac{2\pi}{r}\int \mathrm{d}k\,\frac{k^2}{k^2+1/\ell^2}J_1(kr)\bar{P}(k).
\tag{95}
$$

Interestingly, as we see the drift is not in the radial direction, it is along the $\phi$ direction.

---

[15]Comparing (84) and (85) we learn that (89) may be written in terms of $\Delta u_i$ instead of the charge aspect variation $\Delta\gamma_i$ by simply replacing $\ell$ with the Rossby radius $R$.

[16]Note that $\Delta x^i_{{}_D}\neq -\frac{2}{H}\epsilon^{ij}\partial_j\Lambda(x)$, due to the appearance of $R$ instead of $\ell$ in the argument of the Bessel function $K_0$.

[17]The above is written for $g\neq 0$. For $g=0$ (92) implies $\Lambda=\frac{H}{f}\bar{P}$.

**Hardball example.** For $\bar{P} = P_0\theta(r_0 - r)$, where $\theta$ is the step function,

$$\bar{P}(k) = P_0\int r\,\mathrm{d}r\,\mathrm{d}\phi\;e^{-ikr\cos\phi}\theta(r_0 - r) = 2\pi P_0 r_0^2\frac{J_1(kr_0)}{kr_0}, \tag{96}$$

and hence

$$\Delta x_D^\phi = \frac{fP_0}{gH}\frac{4\pi^2 r_0}{r}\int \mathrm{d}k\;\frac{k}{k^2 + 1/\ell^2}J_1(kr)J_1(kr_0) = \frac{fP_0}{gH}\frac{4\pi^2 r_0}{r}I_1\left(\frac{r_<}{\ell}\right)K_1\left(\frac{r_>}{\ell}\right), \tag{97}$$

where $r_>$ ($r_<$) is the bigger (smaller) of $r, r_0$. Since typically $r, r_0 \ll \ell$ we can expand the Bessel functions to obtain $\Delta x_D^\phi = \frac{\pi fP_0}{4gH}\frac{4\pi r_0^2}{R^2}$. As we see this expression is $r$ independent and is proportional to the area $4\pi r_0^2$ of the hardball over the Rossby radius squared.

**Some comments on our Darwin drift analysis:** (1) As we see from (90) or more explicitly in the hardball example (97), it is linearly proportional to Coriolis parameter $f$. (2) This displacement is essentially transverse to gradient of $P$, e.g. as we see from (95) a radial pressure yields a drift in $\phi$ direction. This "transverse drift" is a result of Coriolis term. (3) It is first order in fluid velocity, unlike the Stokes drift which is second order (cf. discussions of section 6). (4) The usual Darwin drift, e.g. discussed in [16, 17], is however a second order effect and (essentially) independent of $f$. We could have extended our analysis to the second order in the Lagrange formulation, along the same lines discussed in the previous section.

## 8 Discussion

The realization that shallow water system admits a gauge theory description [1] has important interesting physical implications. In particular, in the linearized case, we deal with a Maxwell-Chern-Simons theory which has interesting topological features. The parity violation due to the presence of the Chern-Simons term is a manifestation of the Coriolis term (rotation of the Earth for oceans). Typical electromagnetic wave solutions correspond to the Poincare waves, which manifests this parity violation.

A specific feature of the shallow water system is that it is naturally formulated in a finite spatial span. So, besides the bulk modes (like Poincare waves) we have boundary effects, boundary modes, and boundary degrees of freedom, e.g. Kelvin coastal waves. On the other side, gauge theories in the presence of boundaries and/or their structure in low energy (IR) limit have been under extensive and intensive study in the last decade. These studies have led to a systematic formulation of boundary/asymptotic modes in the context of gauge theories, e.g. see [7], references therein, and its citations. We did not specify boundary/fall off behavior on the gauge fields in our analysis and allowed for generic residual gauge transformations. We discussed that the drifts (fluid side) which correspond to fluid memory effects (gauge theory side), are governed by these boundary modes. It is desirable to study further the boundary modes and their dynamics using the techniques developed in the gauge theory side. The same boundary modes may be relevant to describing topological features of the shallow water systems and/or 2 dimensional condensed matter systems, see e.g. for [1, 5, 6, 39] for recent ideas or studies in this direction.

Moreover, fluid systems can provide a lab to examine and measure the memory effect and its relation to the boundary/asymptotic modes and the infinite set of surface charges. Our fluid memory effects provide a unified framework to interpret Stokes, Darwin, and Euler drifts as displacement memory effects. We showed that the Euler drift is expressed in terms of the memory field. To relate the memory effect to the surface charges, we demonstrated that the

memory field appears as a soft dissipative effect in the non-conservation (change in the value) of the surface charges. We also showed that in the context of the memory setup, a part of the information of a passing wave is encoded in the surface charge associated with large gauge transformations.

The gauge theory description of [1] describes linearized shallow water systems in the Euler formulation. In the Lagrange formulation, as we discussed, the residual/improper part of the $u(1)$ gauge symmetry is replaced by 2 dimensional area-preserving-diffeomorphisms, the latter is the continuum limit of the freedom in labeling the fluid parcels at the surface of the fluid. It is interesting to explore if this symmetry should be promoted to a noncommutative gauge symmetry once we consider very short or very long wave lengths, see [34] for similar ideas.

We close by some other possible future directions.

- It would be interesting to explore fluid memory as holonomy, in either of the Euler or Lagrange formulations. See [40] (and references therein) for related ideas for the case of usual Maxwell or Einstein gravity theories.

- Displacement memory is the leading memory effect in the gravitational systems, see [41] and references therein. We have a similar displacement memory in fluid mechanical systems. It is interesting to relate the two especially in the Lagrange formulation of the fluid where diffeomorphisms appear as the symmetry. In the same direction, there are "analogue gravity" systems and acoustic black holes [42–45] and one can explore bearings of the fluid memory discussed here for acoustic black holes.

- Here we mainly focused on memory in linearized Maxwell-Chern-Simons theory. Given the gauge theory description of the nonlinear theory, one may extend the same analysis to the nonlinear system. The first step towards this has been taken in appendix B. It is interesting to study the interplay of nonlinearity in Stokes drift and the one arising from nonlinear fluid dynamics.

## Acknowledgements

We thank Erfan Esmaeili, Ali Najafi, Blagoje Oblak, Ali Seraj, and David Tong for helpful discussions or comments. We would also like to thank the anonymous referees for their constructive comments which helped improve our presentation. MMShJ acknowledges SarAmadan grant No. ISEF/M/401332.

## A  Green's function

One may rewrite (59) as an equation for the memory field,

$$\left(\partial_i\partial_j - \frac{1}{\ell^2}\delta_{ij}\right)\Delta A_j = -\frac{f}{g}\Delta\gamma_i. \tag{A.1}$$

It is immediately solved using Green's function satisfying

$$\left(\partial_i\partial_k - \frac{1}{\ell^2}\delta_{ik}\right)G_{kj}(x-x') = -\delta_{ij}\delta^2(x-x'). \tag{A.2}$$

To solve equation (61) the standard trick is to go to the Fourier space and back to the real space. We write the Fourier transformation of $G_{ij}(x-x')$ and $\delta(x-x')$ as follows

$$G_{ij}(X) = \int d^2k \bar{G}_{ij}(k)e^{ik\cdot X}, \qquad \delta^2(X) = \frac{1}{(2\pi)^2}\int d^2k e^{ik\cdot X}. \tag{A.3}$$

For brevity, we define $X = x - x'$. Then (61) reduces to

$$\left(k_i k_k + \frac{1}{\ell^2}\delta_{ik}\right)\bar{G}_{kj} = \frac{1}{(2\pi)^2}\delta_{ij}. \tag{A.4}$$

One can simply solve this $2 \times 2$ algebraic equation for $\bar{G}_{ij}$,

$$\bar{G}_{ij} = -\frac{\ell^2}{(2\pi)^2}\left(\frac{k_i k_j}{k^2 + 1/\ell^2} - \delta_{ij}\right). \tag{A.5}$$

Therefore,

$$G_{ij}(X) = \int d^2k\, \bar{G}_{ij}(k)e^{ik\cdot X} = -\frac{\ell^2}{(2\pi)^2}\int d^2k\left(\frac{k_i k_j}{k^2 + 1/\ell^2} - \delta_{ij}\right)e^{ik\cdot X}. \tag{A.6}$$

One can rewrite the Green function as follows

$$G_{ij}(X) = \frac{\ell^2}{(2\pi)^2}\,\partial_i\partial_j I(X) + \ell^2\,\delta_{ij}\delta^2(X), \tag{A.7}$$

where

$$I(X) = \int d^2k\frac{e^{ik\cdot X}}{k^2 + 1/\ell^2} = I(X) = 2\pi K_0(X/\ell), \tag{A.8}$$

where $K_0$ is modified Bessel function of order 0. Recalling that

$$\partial_i\partial_j I(X) = \frac{X_i X_j}{X^2}\partial_X^2 I + \left(\delta_{ij} - \frac{X_i X_j}{X^2}\right)\frac{1}{X}\partial_X I, \tag{A.9}$$

and that

$$\partial_X I = -\frac{2\pi}{\ell}K_1(X/\ell), \qquad \partial_X^2 I = \frac{2\pi}{2\ell^2}\left(K_0(X/\ell) + K_2(X/\ell)\right), \tag{A.10}$$

where $K_1, K_2$ are the modified Bessel functions of order $1, 2$, we arrive at

$$G_{ij}(X) = p(X)\delta_{ij} + q(X)X_i X_j, \tag{A.11}$$

where $X_i = x_i - x'_i$, $X = |x - x'|$,

$$p(X) = \ell^2\delta^2(X_i) - \frac{1}{2\pi}\frac{K_1(X/\ell)}{X/\ell}, \qquad q(X) = -\frac{1}{2\pi X}\partial_X\left(\frac{K_1(X/\ell)}{X/\ell}\right). \tag{A.12}$$

## B  Memory in nonlinear shallow water

In the main text we mainly focused on the charges, memory and the drifts in the linearized shallow water case. The nonlinear theory has also a gauge theory description and hence one may extend the notion of symmetries, charges and memory to this case too, which we briefly discuss here.

**Memory from the equation of motion.**  Nonlinear shallow water equation is

$$\partial_t u_i = -\partial_i\left(\frac{u^2}{2} + gh\right) + \widetilde{E}_i. \tag{B.1}$$

If we fix the temporal gauge for $\widetilde{A}_\mu$, namely $\widetilde{A}_t = 0$, then

$$\partial_t u_i = -\partial_i\left(\frac{u^2}{2} + gh\right) + \partial_t\widetilde{A}_i. \tag{B.2}$$

By integrating over time from both sides, we get

$$\Delta\gamma_i = \Delta\tilde{A}_i - \partial_i \int_{-T}^{+T} \mathrm{d}t \left(\frac{u^2}{2} + gh\right),\tag{B.3}$$

where $\gamma_i = u_i$ is the charge aspect associated with the surface charge (36). We note that temporal tilde-gauge fixing still leaves us with time-independent gauge transformations and that $\Delta\tilde{A}_i$ (like $\Delta A_i$) is invariant under this residual gauge transformations.

**Memory from conservation law.** Similar to the linear case, by using the EoM we can write the conservation of surface charges (36)

$$\Delta\Gamma_\Lambda = \oint_\mathcal{B} \Lambda\Delta u \cdot \mathrm{d}l = \oint_\mathcal{B} \Lambda\Delta\tilde{A} \cdot \mathrm{d}l - \int_{-T}^{+T} \mathrm{d}t \oint_\mathcal{B} \Lambda\nabla\left(\frac{u^2}{2} + gh\right) \cdot \mathrm{d}l \, .\tag{B.4}$$

The above may be understood through definition of $\tilde{B} = f + \zeta$, which yields,

$$\partial_{[i}\tilde{A}_{j]} = \frac{1}{2}f\epsilon_{ij} + \partial_{[i}u_{j]} \quad \implies \quad \Delta u_i = \Delta\tilde{A}_i + \nabla_i X \, ,\tag{B.5}$$

for some $X$ which is undetermined. If we choose $X = -\int_{-T}^{+T} \left(\frac{u^2}{2} + gh\right) \mathrm{d}t$, this equation reproduces (B.4). Note that $X$ in the above may not be thought as a tilde-gauge transformation on $\Delta\tilde{A}_i$, which is gauge invariant, as commented above.

We also note that $\Delta\Gamma_\Lambda$ may also be written in terms of $A$ gauge field, $\Delta\Gamma_\Lambda = \oint_\mathcal{B} \Lambda\epsilon_{ij}E_i/B \, \mathrm{d}l_j$.

The RHS of (B.4) identifies the sources of the non-conservation for $\Gamma_\Lambda$. The first term may be interpreted as the *soft dissipative* term which is an extension of the circulation charge (3) by the insertion of the arbitrary time-independent function $\Lambda$. The last two terms which come with an integral over time are *hard dissipative* terms and have two parts. The first term arises from the nonlinearity of the theory which is analogous to the news function in gravity and the second term plays the role of the external source, which is the counterpart of the matter field in gravitational theories.

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
