# Peer review of "Shallow Water Memory: Stokes and Darwin Drifts"

_SciPost Physics, doi:SciPost Phys. 15, 115 (2023)_

## Round 2 · Referee Report · Anonymous (Referee 1) · 2023-5-29

Strengths

1- The premise of the paper is original and potentially fruitful: it consists in applying modern tools on the global symmetries of gauge theories to a recent gauge-theoretic rewriting of shallow water dynamics in 2+1 dimensions. 2- The result is a beautifully pluridisciplinary work. 3- The computations related to gauge-theoretic symmetries and observables seem to have been carried out carefully. 4- The paper begins with a review of the gauge-theoretic formulation of shallow water, though this also has drawbacks: see the weaknesses below.

Weaknesses

1- The presentation assumes that the reader is familiar with asymptotic symmetries, which is typically not the case for readers coming from condensed matter or hydro communities. This makes the paper very hard to read for most physicists except those that belong to a very specific niche. 2- The review on Tong's gauge-theoretic reformulation of shallow water waves is somewhat superficial: very little value is added compared to Tong's initial paper. In this sense, the effort to present a review is justified and very welcome, but it would have been better if the authors had seized the opportunity to present the material more pedagogically and address the many questions left unanwered in Tong's work. See the requested changes below. 3- The writing style is somewhat amateurish, as if the authors hadn't re-read their draft; this makes the reading more difficult at times. Again, see some requested changes below.

Report

The paper addresses a natural and timely question, using a combination of gauge-theoretic tools (asymptotic symmetries) and recent insights on effective topological gauge theories in hydrodynamics. It definitely deserves to be published in SciPost. However, in view of the weaknesses mentioned above and the scope intended by the authors, several modifications need to be implemented before publication; they are outlined below.

Requested changes

1- In the abstract, the statement that the paper offers a "local generalization of the Kelvin circulation theorem" is puzzling, given that Kelvin's circulation theorem is local to begin with. What the authors have in mind presumably relies on their eqs. (3.4) and (3.11), which contain arbitrary integrands along the boundary of the region $\Sigma$. It would be good to somehow anticipate this in the abstract without using the ambiguous words "local generalization". 2- A general issue of the paper is the use of area-preserving diffeomorphisms as a gauge redundancy, since diffeomorphisms are notoriously a global symmetry of fluids. The same problem affects Tong's approach to hydrodynamics, and is probably very hard to fix; but it would be good if the authors could comment on this, at least to point out that the issue is open. 3- Around eqs. (2.1), the authors state that the Coriolis parameter is constant, and footnote 1 immediately afterwards states that the Coriolis parameter depends on the azimuth $\theta$. The authors should explain how these two statements do not contradict each other -- presumably because the flows they consider take place on small neighbourhoods on a sphere, so that the Coriolis parameter is indeed approximately constant. 4- Below eq. (2.3), what does "generic section of a time slice" mean? Is it a subset of a time slice? If so, why would the integral over $\Sigma$ be conserved? The Noether theorem is normally meant to hold on an entire time slice, not a portion thereof; this should be clarified. Relatedly, what is the surface $\Sigma$ used in eq. (3.5)? Is it a whole spatial slice or just a subset thereof? If it's a subset, how do the authors generalize the notion of asymptotic symmetries for subregions of space? 5- Above eq. (2.5), the shorthand "EoM" was never defined (and the fully written "equations of motion" is surprisingly used, without shorthand, much later in the paper). 6- About the Clebsch parametrization below eq. (2.5): it would be good if the authors could write the action in terms of $\chi,\alpha,\beta$ instead of $\tilde A$, since the latter is ultimately not meant to be varied when computing the equations of motion. In this sense, $\tilde A$ is just a notation for a much more complicated combination of dynamical fields. This may be standard in hydrodynamics, but this is by no means obvious to readers whose background is field theory. Relatedly, the comment at the start of section 3 that "$\tilde A$ is viewed as a dummy field (...)" should definitely also appear the first time $\tilde A$ is introduced. 7- Below eq. (2.15), the citation to Tong is appropriate only to the extent that he wrote something similar -- but in truth, the link between boundary terms in gauge variations and edge modes is known since a much longer time, going back at least to the work of Moore, Seiberg, Witten and Henneaux on Chern-Simons gauge theories, plus work by Wen and others on effective field theories in quantum Hall systems. It would be fair to fix the citation appropriately. 8- At the start of section 2.3, "some different wave solutions" sounds awkward. Replace "some different" by "various"? 9- When talking about Kelvin waves at the end of section 2, the authors surprisingly fail to mention that these are edge modes -- this should definitely be remedied. In addition, from a condensed matter perspective, the choice to impose the sharp boundary condition $u_1 =0$ at the boundary is extremely delicate --- cfr the work by Delplace, Tauber and Venaille on violations of the bulk-edge correspondence. This is not directly related to what the authors discuss, but it is physically crucial, so perhaps the authors could at least comment on that in passing. 10- Above eq. (3.2), the statement that the Noether current is a total derivative is common to all gauge theories; this should be mentioned, since unacquainted readers will otherwise think that this is somehow specific to the model under consideration. 11- Below eq. (3.5), the sentence that goes "As a nomenclature (...) transformations (improper)" is only clear to asymptotic symmetry specialists, and will be completely confusing to other readers. The authors should clarify where the terminology comes from and how it distinguishes global symmetries from genuine gauge redundancies. 12- Is eq. (3.9) the linearization of (3.2)? If yes, this is not manifest at all -- this should be clarified. Relatedly, how is the presence of a central extension in eq. (3.14) compatible with the centreless algebra (3.7) of the nonlinear theory? This is disturbing, at the very least, and the authors say nothing about it -- the omission needs to be remedied. If the origin of the mismatch is understood, the authors need to explain it; and if not, the authors should at least acknowledge that there is a mismatch and state that this is an open problem. 13- Above eq. (4.1), the citation refers to a recent paper by Susskind, but the actual material used in the paper relies entirely on the much earlier paper by Bahcall and Susskind; the citation should be fixed accordingly. 14- At the end of section 4, the authors exhibit area-preserving diffeos as a global symmetry; this is as it should be, since there is no gauge symmetry under diffeos in classical fluids. So what justifies the enhancement of this global symmetry into a gauge redundancy? In what sense are classical fluid parcels indistinguishable, as if they were electrons? Again, this is a deep conceptual issue in Tong's approach to the problem, and it may be very difficult to solve, so it's sufficient if the authors merely point out the problem. 15- The discussion of relevant scales in section 5.1 is illuminating, but it lacks a key ingredient: in what regime of parameters is the shallow water approximation valid to being with? For example, is it clear that it is uniformly valid even when the average depth varies, as is the case close to a beach? If there is a quick answer, it would be nice if the authors could outline it or at least mention some references that address the issue. 16- The sequence of equations (5.5)--(5.8) is a bit mysterious to me. What's the added value? Why are these equations written there? What do they teach us about the (otherwise very nice) memory formula (5.3)? 17- At the very end of section 5, "with waves that these two terms" misses some words. 18- At the start of section 7, why mention Charles G. Darwin, 1953 explicitly in the text? I guess this serves to explain to the reader that this is not quite the same Darwin as in biology; if so, the comment would fit better in a footnote.

  • validity: high
  • significance: good
  • originality: high
  • clarity: ok
  • formatting: reasonable
  • grammar: acceptable

Author:  Vahid Taghiloo  on 2023-06-04  [id 3708]

(in reply to Report 1 on 2023-05-29)

We would like to thank the referee for careful reading of the paper and for constructive comments. In fact, it is some time that we have not seen such detailed, precise, and constructive report on our papers. We have tried to address all of them and improve our paper accordingly. Please see below for more details.

1-We agree with the referee. We have improved the wording of the abstract. What we have is more like a reinterpretation of Kelvin's circulation theorem and a rewriting of it as a conserved quantity.

2-There are two remarks we can make here: (1) Area-preserving diffeomorphisms (APDs) appear in the Lagrange description of the fluid, as we discuss in section 4, following Bahcall-Susskind analysis, whereas what Tong studies is the Euler description. In our understanding APDs do not have a (direct) appearance in the Euler description. (2) The notion of local and global symmetries should be used in a more precise way. In the usual sense, \emph{global} symmetries are generated by some numbers (as opposed to functions) and \emph{local} symmetries are generated by some functions. However, these functions can be generic functions over the whole spacetime \emph{or} be defined only over a subspace of codminesion 1 or 2. The former is what usually known as local gauge symmetries, while the latter is what yields boundary/asymptotic or corner symmetries and surface charges. The ``improper'' gauge transformations are among the latter. With this remark, APDs are in fact among the latter class and are not gauge redundancies. At least classically one can use these symmetries and associated charges to label boundary modes. Recall also that as we discuss in section 4 the APDs are associated with the ``residual gauge symmetries'' once we fix the temporal gauge. We have highlighted this point in the text above eq.(4.8).

3-We have improved the footnote on page 3 to address this point.

4-That is a very good remark. We have tried to fix it by expanding the discussions below eq.(2.3), adding a figure (Fig.1), an equation, (2.4) and some remarks below (3.5).

5-Thank you for catching this.

6-We have revised equation (2.5) and the text below it to address the above.

7-Done!

8-Done!

9-Footnote 6 and a sentence in the one-to-the-last paragraph of section 2 has been added to mention this point.

10-We have mentioned that for any gauge theory, Noether current is the divergence of an anti-symmetric 2-tensor.

11-We have expanded the discussion there and added a reference to Teitelboim et al (1977) and Henneaux-Teitelboim gauge theory book.

12-That is a very good point. Indeed (3.9) is not just a linearization of (3.2). To obtain (3.9) one needs to integrate out the ``dummy gauge field'' $\tilde{A}_\mu$. More explicitly, in the nonlinear theory in principle one has two $u(1)$ currents and (3.2) represents only one of them, and importantly, due to the presence of the CS term these two $u(1)$ currents do not commute. When one integrates out the $\tilde{A}_\mu$ gauge field, one remains with a single $u(1)$ current but now with a non-zero Kac-Moody level, i.e. eq.(3.9).
We have added some comments along the above discussion in the text below (3.9) and (3.14).

13-The citation to Bahcall-Susskind is added.

14-Please see our comments on point 2 above. We have added some comments on this point in the main text at the end of section 4.

15-We have added two sentences in the 2nd paragraph of section 5.1 to make the above more explicit.

16-We have improved the presentation there, along with the required changes in the appendix.

17-Corrected!

18-A footnote has been added on page 19 to provide an explanation.

We have attached a revised version of the paper (in PDF format) in which the changes have been highlighted in red.

We would like to thank again the referee for his/her constructive comments.

Attachment:

Shallow_Water_Memory-colored-changes.pdf

---

## Round 2 · Referee Report · Anonymous (Referee 1) · 2023-6-14

Report

The authors have now addressed the comments raised in my previous report. I believe the paper can now be published as is.

---

## Round 2 · Referee Report · Anonymous (Referee 2) · 2023-7-7

Report

In this article the authors study and push further the gauge theory reformulation of the shallow water system presented by Tong in 2209.10574[hep-th]. In particular, they explain in details the reinterpretation of the so-called Euler/Stokes/Darwin drifts, or fluid memory effects, in terms of large gauge transformations of the gauge theory formulation of shallow water. This is presumably motivated in parts by the developments and results, over the past 10 years, in the study of the infrared asymptotic structure of gauge field theories (such as gravity and electromagnetism). It is indeed in this setup that the relationship between memory effects and large gauge transformations has been made explicit and studied in details. The present work is therefore a very interesting perspective on these ideas. It gives a new and interesting perspective on known results in fluid dynamics, while also providing a new testbed and examples of the memory/symmetry interplay, which could be of interest for the high energy community.

My questions and suggestions for improvements are the following:

1) Could the authors comment briefly on the conserved charge arising from (2.2a), in the same way as (2.3) arises from (2.2b)? At this stage it seems like the two conservation equations (2.2) are on the same footing, while afterwards in the gauge theory description it is clearly (2.2b) which is singled out since (2.3) arises as the charges of the local gauge transformations. This suggests that there might be yet another dual formulation where the charge of (2.2a) arises from gauge transformations.

2) Below (2.3), I believe that at this stage more justification should be needed to conclude from (2.3) that there should be a gauge theory description. After all, codimension-2 integrals appear whenever there is a conserved current as in (2.2), even for a global symmetry.

3) In section 2.1 the authors could perhaps explain in slightly more details which relations between the physical fields $(\mathcal{H},u^i)$ and the electromagnetic fields $(E,B,\tilde{E},\tilde{B})$ are assumed to hold. From what I could understand from reference [1], only the first line of (2.4) is assumed from the onset. This in turn implies, via the Bianchi identity, the conservation of mass, which is therefore already satisfied in this formulation. Then, it seems that from the equations of motion one recovers from the variation with respect to $A_0$ that $\tilde{B}=\xi$, while the variation of the Clebsch parameters $\alpha,\beta$ implies that $\tilde{E}$ is given by (2.4). Then combining these informations from 3 of the EOMs, the 4th EOM (2.6) finally becomes the shallow water equation. This is at least how I understood the construction of [1]. It seems that for the rest of the article these details on the EOMs are not important, but it could help the reader to explain in more precisely which relations between the gauge fields and $(\mathcal{H},u^i)$ are assumed as opposed to derived from the EOMs.

4) Since the authors mention the potential vorticity in (2.9), maybe it could be useful to also explain that before linearization there is also a conserved potential vorticity arising from combining the two equations in (2.2). In particular, the remark below (2.9) about the fact that we can trade the two conservation laws for a single conservation law and the conservation of $Q$, seems to also be true in the non-linear case (although in this case the conservation of $Q$ involves a material derivative).

5) Below (2.11) the authors focus on the sector $Q=0$. It could be useful to state how restrictive or not this choice is, since it leads to the MCS action which is studied at length in the rest of the paper, and no mention of $Q\neq0$ seems to be made later on. In particular, it is not clear whether the discussion of the solutions in section 2.3 are with $Q=0$ or not.

6) In the Kelvin circulation theorem the conservation law is stated with a material derivative, meaning that the contour ($\mathcal{B}$ in equation (3.5)) is advected with the flow. Could the authors comment on how this matches with the conservation of circulation arising from the gauge charge (3.5)?

7) As mentioned in section 2.1, the Clebsch parametrization for the gauge field $\tilde{A}$ is used. This then implies that the gauge transformations on $\tilde{A}$ cannot be implemented anymore in the action (2.5). Yet the authors mention this gauge transformation above (2.6). This requires perhaps a clarification. Also, what is the Noether charge then associated with the sector $\tilde{A}$ if we allow to perform its gauge transformation (i.e. before using the Clebsch parametrization)?

8) The results of section 3.2, although I could reproduce them and appear to be correct, seem a bit counterintuitive. Indeed, it seems that the linearized theory has more structure than the non-linear theory presented in the previous section. In particular $f$ already appears naturally in the charge (3.10), and also as a central term in the algebra (3.14. This seems to suggest that linearization does not commute with the analysis of the symmetries. In other words, why is it not possible to reproduce the results of this section by linearizing the results of section 3.1? Also, by analogy with U(1) gauge theory versus a non-abelian Yang-Mills theory, I would have expected that the non-linear theory has a central extension in its algebra of charges while the linearized theory doesn’t. Similarly, it seems that the results about non-linear memory in appendix B will not reduce to those of section 5 upon linearization.

9) In section 4 the authors explain the relationship between the gauge symmetries, the particle relabelling symmetry and area-preserving diffeomorphisms. I think that some clarifications could be added to answer the following points: - Is it possible to actually turn the theory (2.5) written in Eulerian form into a Lagrangian viewpoint? and if so, what is then the translation of the U(1) gauge transformations of (2.5) from the Lagrangian viewpoint? It seems to me that the authors were heading this way in this section, but didn’t write an action for the model in Lagrangian viewpoint. Is this because of a technical difficulty? - Many authors (see Morrison, 'Hamiltonian description of the ideal fluid’, Shepherd, ‘Symmetries, conservation laws and Hamiltonian structure in geophysical fluid dynamics’, and Salmon, ‘Hamiltonian fluid mechanics’) have shown how the conservation of potential vorticity arises from invariance under particular relabelling symmetry. I fail to see however if this is related to the Lagrangian viewpoint presented by the authors in section 4. Perhaps it could be useful to cite these results on fluid relabelling symmetry.

10) In section 5.2 I have the same comment as above about the role of $Q=0$. What goes wrong if we are outside of this sector?

11) As a general question for sections 6,7,8, can the authors comment on the role of boundary conditions in the fluid systems being discussed? Boundary/fall-off conditions play an important role in the gauge field theory description of asymptotic symmetries, and the subsequent relation with memory effects, while in the present work it seems that the role of boundary conditions has not been discussed (perhaps because they don’t play a role, but then it would be important to point this out).

  • validity: -
  • significance: -
  • originality: -
  • clarity: -
  • formatting: -
  • grammar: -

Author:  Vahid Taghiloo  on 2023-07-13  [id 3806]

(in reply to Report 3 on 2023-07-07)

We would like to thank the referee for careful reading of the paper and for constructive comments. We have tried to address all of them and improve our paper accordingly. Please see below for more details. As a general note, some of the comments from the second referee overlapped with those of the first referee. For this reason, we have made changes related to the first referee in red in the file, and ``new'' suggestions from the second referee have been applied in blue.

1- The conserved charge resulting from equation (2.2a) is simply the mass and its local version is the mass density. We have included a comment in blue in equation (2.2) below.

2-Let us make two comments: (1) (2.2a) may be written as $\partial_\mu J^\mu=0$ and (2.2b) as $\partial_{\mu} \tilde J^\mu=0$. Nonetheless, they are different in the sense that $\tilde{J}^\mu=\partial_\mu f^{\mu\nu}$, for some antisymmetric $f^{\mu\nu}$ which is local in usual fluid variables ${\cal H}, u_i$. One may not do the same for $J^\mu$. Exactly this feature of $\tilde{J}^\mu$ allows one to write its conserved charge as a codimension 2 integral, whereas that is not possible for $J^\mu$. Of course the existence of such an $f^{\mu\nu}$ is the signature of the existence of a gauge symmetry. (2) It is not true that all Noether charges can be written as surface (codimension 2) integrals. If a symmetry is a part of a local (gauge) symmetry then there exists a codimension 2 representation for the Noether charge. A prime example of this is the usual Gauss law.

3-We concur with the referee on this point. In reference [1] a part of the dictionary is assumed to hold and the rest is derived from the variation of the action with respect to different dynamical fields. Although we could have repeated this analysis here, we chose to introduce the entire dictionary at once to keep the review section of the article brief. Besides the comments in red, we have added a sentence above equation (2.5) to refer to Tong’s paper.

4-We thank the referee for bringing this to our attention. David Tong (in the revised version of his paper) also introduced the nonlinear notion of potential vorticity in his paper. In this regard, we have added footnote 4 to refer to Tong’s paper for a nonlinear version of potential vorticity.

5-As the referee noted, our draft primarily focuses on the MCS theory, which is an effective theory for the $Q=0$ portion of the solution space. We did not examine the $Q \neq 0$ case in detail and had only a footnote (footnote 5) on it. We have also included a sentence at the beginning of subsection 2.3 to emphasize that the solution considered in that section is associated with $Q=0$.

6-This is an excellent point, and it was also raised by the first referee. We have added comments in red below equation (3.5) to address this question.

7-The gauge field $\tilde{A}$ is actually a dummy field, and we do not compute the variation of the action with respect to this field. It is the Clebsch parametrization of $\tilde{A}$ which should be used and yields the desired EoM. To clarify this point we have restructured section (2.1), expressing the action in terms of Clebsch fields.

8-This comment is also very relevant. The key point is that the MCS action results from linearization, as well as integrating out $\tilde{A}$ from the action. To address the referee's point, we have added a paragraph in red below equation (3.9) and a comment in red below equation (3.14).

9- In the current draft, we did not include an action for shallow water in the Lagrangian picture. However, references [33,34] present an action for describing an incompressible fluid in $2+1$ dimensions using the Lagrangian picture. In section 4, we interpreted the \textit{residual part of} $U(1)$ gauge symmetry in the temporal gauge in the Eulerian picture as a relabeling symmetry in the Lagrangian picture, as shown in equation (4.8). We have also added a comment below equation (4.8) to clarify this point. Thank you for introducing us to these papers; we have cited them at the beginning of section 4.

10-The $Q\neq 0$ case can be analyzed separately in detail. We have only included some comments in certain parts of the draft to mention this point. Specifically, we have added footnote 12 to explain the changes that occur in equation (5.1) when considering the $Q\neq 0$ case. We hope this comment is enough.

11-As the referee has pointed out in the conventional analysis boundary conditions/fall off behavior seem to play an essential role. However, as we have shown explicitly in some of our recent papers, e.g. 2110.04218 [hep-th] and 2202.12129 [hep-th] even in the gravity case one can bypass this. Let us elaborate a bit. One can construct a largest consistent boundary/asymptotic solution phase space and associate surface charges. Imposing boundary/fall off conditions then amounts to imposing 2nd class constraints on this largest solution space and going to a reduced phase space. This viewpoint is not yet appreciated as much as it should. Imposing the boundary/fall off behavior is a part of the definition of the problem and not a necessity for its consistency. With this general comment, in our case, we do not impose any specific boundary/fall off conditions and keep it as general. (Of course in the class of Kelvin coastal wave solutions, there is a specific choice of boundary conditions.) We have added a comment in section 8 to highlight this point.

We have attached a revised version of the paper (in PDF format) in which the changes have been highlighted in red and blue.

We would like to thank again the referee for his/her constructive comments.

Attachment:

Shallow_Water_Memory-red-blue-colored-changes.pdf

Anonymous on 2023-07-17  [id 3814]

(in reply to Vahid Taghiloo on 2023-07-13 [id 3806])

I thank the authors for their reply to my various questions and comments, and for the improvements and corrections made in the manuscript. I believe this latter can now be published in this form.

---

## Editorial Decision

published